# Characterization of Cellulose from Gagome Kelp and Its Effect on Dough, Gluten, and Starch as Novel Bread Improvers

**DOI:** 10.3390/foods14071246

**Published:** 2025-04-02

**Authors:** Xiang Li, Heqi Yang, Xiaohui Yu, Ying Tuo, Hui Zhou, Yidi Cai, Long Wu

**Affiliations:** 1College of Food Science and Engineering, Dalian Ocean University, Dalian 116023, China; lx910702@163.com (X.L.); yanghhhqi@163.com (H.Y.); 15754028359@163.com (X.Y.); ty15842784797@163.com (Y.T.); zhouhui@dlou.edu.cn (H.Z.); caiyidi@dlou.edu.cn (Y.C.); 2Dalian Jinshiwan Laboratory, Dalian 116034, China; 3National R&D Branch Center for Seaweed Processing, Dalian Ocean University, Dalian 116023, China

**Keywords:** cellulose, bread, texture, storage stability, sensory characterization

## Abstract

Novel bread formulations with natural improvers have become an essential part of improving the quality of bakery products. In the present study, novel bread improvers made using Gagome kelp cellulose (GC) were systemically evaluated, and cellulose-improved dough (GC-dough), gluten (GC-gluten), and gluten protein and starch (GC-starch) were all studied. The results indicated that the water and oil holding capacity, cholesterol-adsorptive capacity, and the unsaturated fat and saturated fat adsorptive capacities of GC had increased. GC also showed high glucose adsorptive capacity, antioxidant activity, α-amylase inhibition, and glucose diffusion inhibition activity. Furthermore, the color of the GC-dough was improved with the addition of the GC, which also affected the content of glutenin, the water holding capacity in GC-gluten, and the solubility of GC-starch. In addition, the cross-linked network formed by GC could be observed in the GC-bread, indicating an improvement in texture and sensory evaluation. Bread with 1% (*m*/*m*) added GC provided the highest sensory characteristics and the best cold storage stability, which suggests that it is the best strategy for further study. The results might show a potential application of by-products of marine origin in commercial bakery production.

## 1. Introduction

Bread is known for its bland flavor and spongy crumb, characteristics that have been preserved throughout history. However, technological difficulties frequently appear in improving the nutrition and function of bread. An unexpected appearance and texture may be generated with the addition of healthier ingredients to the bread formulation, resulting in unconventional product characteristics [1,2]. Decreasing the content of gluten proteins might weaken the development of the gluten network in bread when replacing wheat flour with other components [3]. Moreover, the air retention and porosity of the bread dough might be negatively affected by the addition of non-gluten ingredients [4,5,6]. The addition of non-wheat ingredients to the bread formulation might lead to drastic modifications to texture, accompanied by a reduction in quality [7,8,9]. Therefore, the present study proposes the development of a novel functional bread formulation that maintains quality and texture while enhancing nutritional value and offers a viable strategy for improving the health benefits of bread [10].

Among the different bread improvers, cellulose has been addressed in diverse research studies due to its health effects [11]. As a dietary fiber source in the human diet, cellulose is an abundant and sustainable material for functional foods [12]. The bland flavor, white color, and safety of cellulose might be beneficial to the product features of bread. Moreover, increased viscosity could appear in starch-containing foods with added cellulose, limiting the absorption and diffusion of glucose [13]. Furthermore, decreasing the activity of digestive enzymes might be induced by cellulose, which may present diffusional barriers to the contact of the enzymes with the substrate [12]. Therefore, a wide array of cellulose ingredients has been applied in cookies and bread with the purpose of increasing their fiber content or reducing their calorific value [12]. Plants and wood are normal sources of cellulose. However, little research has been conducted with regard to the utilization of algal cellulose.

As early as 1885, Stanford [14] proposed the concept of algal cellulose, which is isolated from the insoluble residue after the extraction of alginates from brown algae. In one of the reports, cellulose from brown algae exhibited superior thickening behavior in protein-rich food through hydrogen bonds, which form a weak gel-like structure [14,15]. Thus, a worthwhile endeavor would be to clarify the potential effects of algal cellulose as a dietary fiber source in the human diet, especially for sensory characterization of the food.

This study focuses on the characterization of cellulose isolated from Gagome kelp (GC), a brown seaweed with significant annual harvest and economic potential, and its application as a novel bread improver [16]. The effects of GC on dough (GC-dough), gluten (GC-gluten), and starch (GC-starch) properties were systematically investigated. Additionally, GC was incorporated into bread formulations to evaluate its impact on morphology, texture, sensory characteristics, and cold storage stability. The results provide a foundation for developing functional, texturally enhanced, and sensorially improved bread formulations using marine-derived cellulose. To the best of our knowledge, this is the first study to comprehensively evaluate the quality of GC-bread derived from Gagome kelp.

## 2. Materials and Methods

### 2.1. Materials

Gagome kelp (*Kjellmaniella crassifolia*), cultivated in the Yellow Sea near Dalian, China, and harvested in May 2020, was provided by a local seaweed farm. The obtained Gagome kelp contained 71.86 (0.21)% water (% RSD values are given in parentheses). The dry basis of raw materials contained 1.73 (1.16)% ash, 0.49 (2.04)% crude protein, and 0.16 (3.50)% mannitol; alginate was not detected in the present samples. Microcrystalline cellulose (MCC) was purchased from Sigma Co., Ltd. (Alexandria, VA, America).

Phthalic acid, sulfuric acid, standard cholesterol solution, glucose, α-amylase, hydrochloric acid, silver nitrate, sodium chloride, and sodium hydroxide were all analytical-grade and were purchased from Macklin Co., Ltd. (Shanghai, China). The ethanol, 2,2-diphenyl-1-picryhydrazyl (DPPH), L-ascorbic acid, ferroferric oxide, salicylic acid, hydrogen peroxide, potassium iodide, and potassium bromide used in the present study were all analytical-grade and were purchased from Sigma-Aldrich Co., Ltd. (Alexandria, VA, America). The ab65333 glucose determination kit was purchased from Abcam Co., Ltd. (Shanghai, China).

In addition, peanut oil was bought from Luhua Cereals & Oils Industry Group Co. Ltd. Potato starch and high-gluten flour were purchased from Lam Soon Co., Ltd. (Hong Kong, China). Yeast was purchased from Lesaffre Co., Ltd. (Marseille, France). Salt was purchased from China National Salt Industry Corporation. Sugar was purchased from Taikoo Sugar Co., Ltd. (Hong Kong, China). Butter was purchased from Lactalis Co., Ltd. (Laval, France). The pig fat and fresh egg yolk were bought from the local market in Dalian, China.

### 2.2. Methods

#### 2.2.1. Determination on Water Holding Capacity of the GC

Firstly, the GC was isolated from the seaweed according to a previously reported procedure [17]. The wet sample was then rinsed with t-butanol, freeze-dried, and stored in a desiccator for further experiments.

For the determination of water holding capacity, 40 mL deionized water was added to 1.0 g cellulose sample; the mixture was stored for 24 h at 25 °C. Then, centrifugation was performed for 30 min at 4000 r/min, and the precipitate was dried until constant weight at 105 °C. Finally, the water holding capacity was calculated by the following Formula (1):(1)$$\mathrm{Water}\;\mathrm{holding}\;\mathrm{capacity}\;(g/g)=\frac{m_{a}-m_{b}}{m_{b}}$$
where *m_a_* represents the weight of precipitate after centrifugation (g); *m_b_* represents the final weight after drying (g). All measurements in the present determination were performed ten times.

#### 2.2.2. Determination of Oil Holding Capacity of the GC

For the determination of oil holding capacity, a 150 mm quantitative filter paper was soaked in peanut oil for 20 min and hung for 30 min until constant weight. Then, 0.2 g of cellulose samples was wrapped with the present quantitative filter paper, and soaked in peanut oil for 20 min. Thereafter, the hanging process was performed for 30 min until constant weight. The oil holding capacity was calculated by the following Formula (2):(2)$$\mathrm{Oil}\;\mathrm{holding}\;\mathrm{capacity}\;(g/g)=\frac{{m\times m}_{a}\times m_{b}}{m}$$
where *m* represents the weight of the cellulose sample (g); *m_a_* represents the weight of quantitative filter paper adsorbed peanut oil (g); *m_b_* represents the total weight of the cellulose sample wrapped with quantitative filter paper (g). All measurements in the present determination were performed ten times.

#### 2.2.3. Determination on Unsaturated Fat and Saturated Fat-Adsorptive Capacity of GC

For the determination of adsorptive capacity of unsaturated fat, 3.0 g of cellulose samples was added to 24 g peanut oil, and the mixture was stored at 37 °C for 2 h. Thereafter, the mixture was centrifuged for 15 min at 4000 r/min. The dissociated peanut oil upon the precipitate was removed by filter paper. Then, the adsorptive capacity of unsaturated fat was calculated by the following Formula (3):(3)$$\mathrm{Adsorptive}\;\mathrm{capacity}\;\mathrm{on}\;\mathrm{unsaturated}\;\mathrm{fat}\;(g/g)=\frac{m_{au}-m_{c}}{m_{c}}$$
where *m_c_* represents the weight of the cellulose sample (g), and *m_au_* represents the weight of the cellulose sample adsorbed peanut oil (g).

For the determination of the adsorptive capacity of saturated fat, peanut oil was replaced by pig fat. After the same storage and centrifugation process, the adsorptive capacity on saturated fat of the cellulose samples was calculated by the following Formula (4):(4)$$\mathrm{Adsorptive}\;\mathrm{capacity}\;\mathrm{on}\;\mathrm{saturated}\;\mathrm{fat}\;(g/g)=\frac{m_{as}-m_{c}}{m_{c}}$$
where *m_au_* represents the weight of cellulose sample adsorbed pig fat (g). All measurements in the present determination were performed ten times.

#### 2.2.4. Determination on Cholesterol Adsorptive Capacity of the GC

The cholesterol adsorptive capacity was determined by the method reported by Angioloni et al. [18]. Briefly, fresh egg yolk was diluted ten times with deionized water, and homogenized into the emulsion. Then, 2.0 g cellulose samples were added to 50 mL egg yolk emulsion. The pH value of the mixture was adjusted to 2.0 and 7.0 with acetic acid, respectively. After storage for two hours at 37 °C, the mixture was centrifuged for 20 min at 4000 r/min. Five milliliters of supernatant were mixed with fifteen milliliters of 0.1 mg/mL phthalic acid solution and ten milliliters of sulfuric acid, and the mixture was stored at room temperature for 10 min. Finally, the absorbance value of the mixture was recorded at 505 nm, and the standard cholesterol solution was performed for the calibration. The cholesterol adsorptive capacity was calculated by the following Formula (5):(5)$$\mathrm{Cholesterol}\;\mathrm{adsorptive}\;\mathrm{capacity}=\frac{m_{1}-m_{2}}{m}$$
where *m*_1_ represents the cholesterol weight of egg yolk emulsion (mg); *m*_2_ represents the cholesterol weight of supernatant (mg); *m* represents the weight of cellulose sample (g). All measurements in the present determination were performed ten times.

#### 2.2.5. Determination on α-Amylase Inhibition Activity of the GC

In order to evaluate the α-amylase inhibition activity of the cellulose, 40 g potato starch was added into 1000 mL 50 mM phosphate buffer (pH 6.5). After stirring for 30 min at 65 °C, 1 g cellulose sample and 4 mg α-amylase were added to the potato starch solution. The mixture was shaken for 1 h at 37 °C, and centrifuged for 20 min at 3000 r/min. The α-amylase inhibition activity of the cellulose samples was calculated by the following Formula (6):(6)$$\alpha \mathrm{-amylase}\;\mathrm{inhibition}\;\mathrm{activity}\;(\%)=\frac{A-A_{1}}{A}$$
where *A* represents the absorbance value of the blank; *A*_1_ represents the absorbance value of the cellulose sample. All measurements in the present determination were performed ten times.

#### 2.2.6. Determination on Glucose Adsorptive Capacity of the GC

For the determination of glucose adsorptive capacity, 1.0 g cellulose samples were added into 100 mL 5, 10, 50 and 100 mM glucose solution, respectively. The mixture was stirred for 6 h at 37 °C, and centrifuged for 20 min *n* at 4000 r/min. Then, the absorbance value of the supernatant was recorded at 505 nm, and the standard glucose solution was performed for calibration. The glucose adsorptive capacity was calculated by the following Formula (7):(7)$$\mathrm{Glucose}\;\mathrm{adsorptive}\;\mathrm{capacity}=\frac{{(C}_{0}-C_{1})\times 0.1}{m}$$
where *C*_0_ represents the glucose concentration of glucose solution (mM); *C*_1_ represents the glucose concentration of supernatant (mM); *m* represents the weight of cellulose sample (g). All measurements in the present determination were performed ten times.

#### 2.2.7. Determination on Glucose Diffusion Capacity and the Glucose Dialysis Retardation Index (GDRI) of the GC

Before the determination of the glucose diffusion capacity, the cellulose samples were washed twice with ethanol to remove the soluble components [12]. Then, 0.5 g washed cellulose samples were added into 15 mL 100 mM glucose solution. After stirring for 1 h, the mixture was removed to a dialysis bag (Mw = 1000) [19]. In order to stimulate intestine peristalsis, the dialysis bag was shaken in 200 mL deionized water for 3 h min at 37 °C, and the shaking process was conducted for 3 h after a 3 h rest.

For the determination of the GDRI value, the glucose content at 30, 60 and 90 min was recorded by the glucose determination kit. Finally, the glucose dialysis retardation index (GDRI) was calculated by the following Formula (8):(8)$$\mathrm{GDRI}\;(\%)=(1-\frac{C-C_{d}}{C_{0}})\times 100\%$$
where *C* represents the glucose concentration of the cellulose sample (mM); *C_d_* represents the glucose concentration of the blank; *C*_0_ represents the glucose concentration of the initial glucose solution.

In addition, for the determination of glucose diffusion capacity, the glucose content at 30, 60, 90, 120, 180, 240, 360, 540, 720 and 1440 min was determined by a glucose determination kit. All measurements in the present determination were performed six times.

#### 2.2.8. Determination on Cation Exchange Capacity of the GC

Before the determination of the cation exchange capacity, the cellulose samples were soaked in 30 mL 0.1 mM hydrochloric acid solution for 24 h, and the sample was washed with deionized water to pH 7.0. Then, the cellulose samples were titrated with 10% (m/v) silver nitrate solution until there was no white precipitation, and that precipitation was collected and dried. Next, 100 mL 0.15 g/mL sodium chloride solution was mixed with 0.25 g dried cellulose sample. The mixture was titrated with 0.1 M sodium hydroxide solution (0.2 mL for each time) until pH remained unchanged. The cation exchange capacity of the cellulose samples was calculated by the following Formula (9):(9)$$\mathrm{Cation-exchange}\;\mathrm{capacity}\;(\mathrm{mmol}/g)=\frac{C\times (V_{1}-V_{0})}{m}$$
where *C* represents the pH value at the end of titration; *V*_1_ represents the volume of sodium hydroxide solution in the titration on cellulose sample (mL); *V*_0_ represents the volume of sodium hydroxide solution in the titration on the blank (mL); m represents the weight of cellulose sample (g). All measurements in the present determination were performed six times.

#### 2.2.9. Determination on DPPH Free-Radical Scavenging Activity of the GC

The determination of DPPH free-radical scavenging activity of the cellulose samples was performed as described by Li et al. [20] with minor modifications. Briefly, 2.0 g cellulose samples were dissolved in 30 mL 70% ethanol, and then 2 mL of the mixture was added to 2 mL DPPH solution (0.4 mM). The mixture was centrifuged at 5000 r/min for 10 min after 30 min of incubation in darkness, and the absorbance of the supernatant was detected at 517 nm. The L-ascorbic acid was used as a control. The DPPH radical scavenging activity (%) was determined using the following Equation (10):(10)$$\mathrm{DPPH}\;\mathrm{scavenging}\;\mathrm{activity}\;(\%)=\frac{A_{b}\times (A_{s}-A_{c})}{A_{b}}\times 100\%$$
where *A_b_* represents the absorbance of the blank; *A_s_* represents the absorbance of the cellulose sample; *A_c_* represents the absorbance of the control.

#### 2.2.10. Determination on Hydroxyl Radical Scavenging Activity of the GC

For the determination of hydroxyl radical scavenging activity, 2.0 g cellulose samples were dissolved in 30 mL 70% ethanol. Then, 0.2 mL cellulose ethanol solution was mixed with 1 mL 0.15 mM ferroferric oxide solution, 0.4 mL 2 mM salicylic acid solution, 1 mL 6 mM hydrogen peroxide solution and 0.4 mL deionized water. After 30 min of incubation at 37 °C, the mixture was centrifuged for 5 min at 5000 r/min, and the absorbance of the supernatant was detected at 510 nm. The L-ascorbic acid was used as a control. The hydroxyl radical scavenging activity (%) was determined using the following Equation (11):(11)$$\mathrm{Hydroxyl}\;\mathrm{radical}\;\mathrm{scavenging}\;\mathrm{activity}\;(\%)=\frac{A_{b}\times (A_{s}-A_{c})}{A_{b}}\times 100\%$$
where *A_b_* represents the absorbance of the blank; *A_s_* represents the absorbance of the cellulose sample; *A_c_* represents the absorbance of the control.

#### 2.2.11. Determination on the Volume, pH and Color of the GC-Dough

During the preparation of the GC-bread, the properties of the GC-dough, GC-gluten and the GC-starch were evaluated. Firstly, the GC-dough was prepared with the GC and flour. Briefly, 2.00 g cellulose samples were mixed with 18.00 g high-gluten flour. Then, 5.9 mL 20 mg/mL sodium chloride solution was added into the mixture with continuous stirring, until the GC-dough was obtained. The raw GC-dough was proofed at 40 °C with 80% humidity for 20, 40 and 60 min, respectively, and the volume of the final GC-dough was determined. Then, the proofed GC-dough was added to 90 mL of deionized water, and the pH of the mixture was determined. The same preparation was performed to obtain the dough added MCC (MCC-dough).

In addition, the quantitative evaluation of the color changes in GC-dough and MCC-dough samples was performed using a colorimeter (HunterLab D65 UltraScan PRO, Reston, VA, USA). The colorimeter was calibrated against a standard calibration plate with a white surface and set to CIE Standard Illuminant C before each measurement series. The color brightness coordinate *L** (from white at 0 to black at 100) and the chromaticity coordinates *a** (green when positive and red when negative) and *b** (yellow when positive and blue when negative) were measured [19]. The assay was repeated three times. Then, the color difference (ΔE) of the dough was determined using the following Equation (12):(12)$$E=\sqrt{\left(L^{*}+a^{*}+b^{*}\right)}$$

#### 2.2.12. Separation of the GC-Gluten and GC-Starch

The separation of GC-gluten and GC-starch was performed as described by Day et al. [21]. Firstly, 50 g of the GC-dough was fermented for 30 min at 25 °C. Then, the GC-dough was washed with 20 mg/mL sodium chloride solution, and the wet GC-gluten obtained was detected by potassium iodide solution until no change of color could be observed. Additionally, the eluate was collected and sieved by a 200 mesh, and the filtrate was centrifuged for 30 min at 5000 r/min. After drying for 24 h at 55 °C, the precipitate was sieved by a 100 mesh to obtain the GC-starch. The same preparation was performed to obtain the gluten (MCC-gluten) and starch (MCC-starch) added MCC.

#### 2.2.13. Determination of the Content and Water Holding Capacity of GC-Gluten

The wet GC-gluten was separated from 50 g of the GC-dough as described in Section 2.2.12. For the determination of the gluten content, the wet GC-gluten was centrifuged at 6000 r/min for 10 min. The precipitation gluten was weighed as *m*_1._ The gluten content (%, C) in the dough was determined using the following Equation (13):(13)$$C\;(\%)=\frac{m_{1}}{m_{d}}\times 100\%$$
where *m*_d_ represents the weight of GC-dough (50 g).

After the determination of the content of GC-gluten, the wet GC-gluten was freeze-dried. Then, the water holding capacity of freeze-dried GC-gluten samples was evaluated with the method described in Section 2.2.1. The same preparation was performed with MCC.

#### 2.2.14. The Secondary Structure of GC-Gluten Protein

The secondary structure of gluten protein was determined by Fourier transform infrared (FTIR) spectroscopy. Before the determination, 1 mg freeze-dried GC-dough and MCC-dough powder was ground and sieved with an 80-mesh sieve, the powder was mixed with 0.1 g potassium bromide powder, and the mixture was pressed into cylindrical tables. The FTIR spectroscopy was recorded by a Fourier transform infrared spectrometer (Frontier, PerkinElmer, Waltham, MA, USA) with a wavelength of 400~4000 cm^−1^.

#### 2.2.15. Determination on the Content of Glutenin and Gliadin

For the determination of the content of glutenin and gliadin, 5.00 g freeze-dried GC-dough and MCC-dough powder was dissolved in 75 mL 70% ethanol solution. The mixture was centrifuged for 10 min at 5000 r/min after 2 h of magnetic stirring. The precipitate was added into 75 mL 70% ethanol solution, and the same stirring and centrifugation process was performed. After washing with deionized water, the precipitate was freeze-dried to obtain the glutenin. At the same time, both supernatants in the centrifugation were collected, followed by vacuum concentrating and freeze-drying to obtain the gliadin.

#### 2.2.16. Determination on the Solubility and Swelling of GC-Starch

Twenty-five milliliters of the deionized water were added into 0.5 g GC-starch and MCC-starch samples, and boiled for 20 min. After cooling to room temperature in an ice bath, the mixture was centrifuged for 15 min at 3000 r/min. The supernatant and precipitate were dried, respectively, until no change in weight could be observed. The solubility (14) and swelling (15) of the GC-starch sample were determined using the following equation:(14)$$\mathrm{Solubility}\;(\%)=\frac{m_{s}}{m_{i}}\times 100\%$$(15)$$\mathrm{Swelling}\;(\%)=\frac{m_{p}}{m_{i}*(100-s)}\times 100\%$$
where *m_s_* represents the starch weight of supernatant (g); *m_i_* represents the initial weight (0.5 g) of starch (g); *m_p_* represents the starch weight of precipitate (g); *s* represents the solubility of the GC-starch sample determined using the equation (14, %).

#### 2.2.17. Preparation of the GC-Bread

For the preparation of the GC-bread powder, the cellulose samples were added into high-gluten flour with the ratios of 0, 1%, 3%, 5% and 7% (m/m, calculated with the weight of high-gluten flour), respectively. Then, 165 g water and 4 g yeast were added to 280 g of GC-bread powder, mixing with a mixer (E-1063, Shunran, Jiangmen, China) for 2 min at 1300 W. Next, 3 g salt and 30 g sugar were added to the mixture. After 2 min of mixing, 25 g butter was added to the mixture, followed by 12 min of mixing. After 20 min of fermentation at 30 °C, the final mixture was divided into 50 g of dough, and those doughs were stored for 15 min at room temperature. The second fermentation was performed for 30 min at 40 °C with 80% humidity. Finally, the doughs were baked in an oven (DKX-C20M3, Xiaoxiong, Foshan, China) for 15 min at 180 °C (top temperature)/200 °C (bottom temperature).

#### 2.2.18. The Scanning Electron Microscopy (SEM) Observation of the GC-Bread

The morphology of the GC-bread was observed by SEM (SU1510, HITACHI, Tokyo, Japan). The freeze-dried GC-bread (cellulose content of 1%, 3%, 5% and 7%) samples were ground, and the powder was put on a specimen stage equipped with conductive adhesive. After coating with a gold-palladium alloy for 60 s at 10 mA, the GC-bread samples were observed by SEM. The images were processed using SEM series software (Tokyo, HITACHI).

#### 2.2.19. Determination on Texture Profile of the GC-Bread

The texture of GC-bread (cellulose content of 1%, 3%, 5% and 7%) was determined as described by Li et al. [22] with minor modifications. For the texture profile analysis (TPA), the top and bottom of the fresh GC-bread were removed, and the center of GC-bread was cut into a 3.0 cm × 3.0 cm × 3.0 cm sample. The TPA was performed by a texture analyzer with a P/0.5S probe (TA-XT2i plus C, Stable Micro Systems, Surrey, UK). Then, the determination was performed for a 2-cycle sequence with 1.5 N strain, 50 mm/min test speed and 25% deformation. The hardness, adhesion, cohesiveness, springiness, adhesiveness, and chewiness of the samples were calculated by the software (TLPro 1.13-002, Stable Micro Systems, Surrey, UK). The measurements were performed ten times, and the results were expressed as mean ± std.

#### 2.2.20. Sensory Test Description of the GC-Bread

A quantitative descriptive analysis (QDA) of GC-bread was tested using a 10-point descriptive profile analysis (Appendix A), as described in Akyüz et al. [23]. The GC-breads samples named 1–5 were cut into 2 cm × 2 cm × 2 cm pieces before being given to the assessors. The test was performed in three sessions by a trained panel of ten assessors (five females and five males, aged 25–50 years) with prior experience in evaluating bakery products. Each session was conducted in standardized sensory booths under controlled conditions. Assessors were provided with water to cleanse their palates between samples. Protocols for the sensory evaluation were approved by the Experimental Ethics Committee of Dalian Ocean University (ethical approval No. 2024102102, approval date: 21 October 2024) and complied with the guidelines. Informed consent was obtained from all participants prior to the study.

#### 2.2.21. Determination on the Moisture Content and Hardness of the Cold-Stored GC-Bread

The moisture content of the GC-bread was determined as described by the Chinese standard GB/T 5009.3 [24]. Briefly, the GC-bread was ground into a powder. Then, five grams of the bread powder were put into a weighing bottle, and dried at constant weight in a 105 °C hot-air drier (FD-S56, Binder, Neckarsulm, Germany). And the moisture content (M) of the GC-bread was canulated using the following Equation (16):(16)$$M=\frac{m_{1}-m_{2}}{m_{1}-m_{3}}\times 100\%$$
where *m*_1_ represents the weight of the weighing bottle and bread sample (g); *m*_2_ represents the weight of the weighing bottle and bread sample after drying (g); *m*_3_ represents the weight of the weighing bottle (g).

The hardness of the cold-stored GC-bread was determined as described by Arp et al. [25]. The GC-bread was stored for 24, 72, 120 and 168 h at 4 °C, and then the same TPA was performed. The hardness of stored GC-bread was fit into the Avrami Formula (17):F_s_ − F_0_ = 1 − exp (−*k*t*^n^*)(17)
where F_s_ represents the hardness of the stored GC-bread sample (N); F_0_ represents the hardness of the fresh GC-bread without storage (N); *k* represents the rate constant; t represents the store time (d); *n* represents the Avrami index.

#### 2.2.22. Statistical Analysis

The mean and standard deviation of the data were calculated for each treatment. Analysis of variance (ANOVA) was carried out to determine any significant differences (*p* < 0.05) among the applied treatments with the SPSS software package (SPSS 16.0 for Windows).

## 3. Results and Discussion

### 3.1. Water and Oil Holding Capacity of the GC

The water and oil holding capacity of GC would affect the water availability of the protein in bread, and delay the development of gluten. As shown in Table 1, the water and oil holding capacities of GC were significantly (*p* < 0.05) higher than those of MCC, suggesting an improved texture of bread with added GC. The results might be attributed to the porous structure of GC, which could adsorb the water and oil droplets in a cross-linking network by steric hindrance [26]. In addition, the hydrophilic hydroxyl group of the cellulose might also increase the water holding capacity of the GC [26].

### 3.2. Interaction Between the GC and Fat

The fat-lowering effect of the cellulose has been widely reported in recent years. For the illumination of the interaction between the GC and fat, the adsorptive capacity of the GC on unsaturated fat and saturated fat was evaluated. As shown in Table 1, the saturated fat adsorptive capacity of the GC was 1.73 ± 0.20 mg/g, which was significantly (*p* < 0.05) higher than those of MCC.

Moreover, the adsorptive capacity of saturated fat of the GC and MCC was higher than the results on unsaturated fat (Table 1), and the steric hindrance induced by a double-bonded electronic cloud of the unsaturated fat might be attributed to the results [27]. The steric hindrance is reduced with the reduced diameter of the unsaturated fat molecule, and weakening the unsaturated fat adsorptive capacity of the present cellulose sample.

### 3.3. Cholesterol Adsorptive Capacity of the GC

Cellulose enhancement of food could be conducive to the formation of mucous on the inner wall of the small intestine, which might be attributed to the cholesterol-lowering effect in vivo [28]. In addition, the network cross-linking by cellulose might also be a barrier layer to limit the absorption of cholic acid, which could increase the consumption of cholesterol. In order to evaluate the cholesterol-lowering effect in vitro, the cholesterol adsorptive capacity of the GC was determined. The results showed an obvious effect on the adsorptive capacity of the GC (Table 1). The cholesterol adsorptive value of the GC at pH 7.0 (5.47 ± 0.38 mg/g) was higher than those of pH 2.0 (4.48 ± 0.14 mg/g), indicating the improved cholesterol adsorptive capacity in the small intestine rather than stomach [28].

### 3.4. The α-Amylase Inhibition Activity of the GC

α-amylase was found in saliva and pancreatic juice, hydrolyzing starch into maltose [12]. Recent studies showed that the α-amylase inhibition activity varied depending on the cellulose sources, concentration and purity [12]. Results in Table 1 indicated the efficient α-amylase inhibition activity of the GC and MCC. Although lower than the inhibition activity of MCC, the GC showed 14.31 ± 0.21% of α-amylase inhibition activity. Such results were supported by the reports of Liu et al. [12], who indicated that a diet including 20% purified cellulose showed efficient α-amylase activity in vivo, and suggested the hypoglycemic effect of cellulose-enhanced food.

### 3.5. Interaction Between the GC and Glucose

Cellulose has potential application in food systems as dietary fiber, which could limit the absorption and diffusion of glucose [12]. For the illumination of the interaction between the GC and glucose, the glucose adsorptive capacity, GDRI and diffusion capacity were determined. The GC showed a concentration-dependent glucose adsorptive capacity, as shown in Figure 1A. The glucose adsorptive capacity of the present GC ranged from 1.23 ± 0.02 mM to 7.27 ± 0.37 mM, increasing with the increased content of glucose (from 5 to 100 mM). Moreover, the present GC showed a stronger glucose adsorptive capacity than the MCC. When the content of glucose increased to 100 mM, the glucose adsorptive capacity value of the GC was 3 times higher than that of MCC. Similar trends have been observed for cellulose-rich fractions from citrus peel [29], which suggested the saturated concentration of glucose was nearly 324 μmol/g. The large surface area of cellulose induced by the cross-linking network might be attributed to such high glucose adsorptive capacity, and increasing the binding force on glucose molecules with high steric hindrance [29]. Furthermore, the high steric hindrance of cellulose might inhibit the diffusion of glucose in vivo [29].

The GDRI value had been used as an indicator of the retardation effect of the cellulose on glucose absorption in the jejunum [12]. As shown in Figure 1B, the GDRI value of the GC showed the highest GDRI value of 14.18 ± 0.04% after 30 min dialysis, significantly (*p* < 0.05) reducing with the increasing dialysis time. Additionally, the MCC showed the highest GDRI value of 15.89 ± 0.13% after 60 min dialysis. The results indicated that both GC and MCC showed an obvious effect on the inhibition of glucose diffusion, which was supported by the results of Figure 1A. Several studies had reported the glucose adsorptive capacity of the cellulose and the retardation of the glucose absorption in vivo by three main mechanisms [30,31]. Firstly, cellulose would increase the viscosity of liquids in the digestive system, and inhibit the diffusion of glucose in the intestinal lumen. Secondly, large molecules in cellulose could bind with glucose, limiting the diffusion which is needed for facilitated transport in the intestine. Finally, dietary fiber may inhibit the activity of α-amylase enzymes that convert starch to glucose. Such mechanisms were in agreement with the results of the present GC, suggesting that the GC exerts hypoglycemic potential attributed to the glucose adsorptive capacity and diffusion inhibition capacity.

Results of Figure 1C showed that the addition of the GC contributes to reduced glucose diffusion. When the diffusion time increased to 240 min, the glucose concentration of the GC was lower than that of the blank control, suggesting the glucose diffusion capacity of the GC, which was in agreement with the high GDRI value. In addition, no significant difference in glucose diffusion capacity was observed between the GC and MCC. The results were similar to the study of rice bran cellulose on glucose diffusion reported by Qi et al. [30], who studied glucose diffusion in rice bran cellulose over 12 h, and indicated that maximum glucose levels were reached after 300 min.

### 3.6. Cation Exchange Capacity of the GC

Acidulous cations, such as Ca^2+^, Zn^2+^, Cu^2+^ and Pb^2+^, could affect the hydroxyl and carboxyl groups of cellulose molecules. The results might suggest that the gastrointestinal buffering capacity might be induced by the cellulose in food, which improved the osmotic pressure in vivo. As shown in Figure 2A, the high cation exchange capacity could be observed in the present GC. The pH value of the GC mixture increased with the addition of the NaOH solution, reaching saturation at pH 9.78. The results showed that cation exchange capacity of GC and MCC was 383.35 ± 10.89 mmol/g cellulose and 147.41 ± 4.88 mmol/g cellulose, respectively.

### 3.7. Antioxidant Activity of the GC

The antioxidant activity of the GC was evaluated by the DPPH and hydroxyl radical scavenging activity. As shown in Figure 2B, the DPPH radical scavenging activity of the GC (67.93 ± 0.19%) was obviously higher than that of MCC (15.66 ± 0.09%).

In addition, the high antioxidant activity of the GC might be in agreement with the high hydroxyl radical scavenging activity, which was related to senescence and peroxide metabolism in vivo. The hydroxyl radical scavenging activity of GC was 78.62 ± 1.92%, and those of the MCC (42.18 ± 0.59%) were obviously lower than GC.

### 3.8. Effect of the GC on Dough of Bread

In order to evaluate the effect of GC on dough characterization, the volume, pH value and color of the GC-dough with different proofing times were determined. As shown in Table 2, the significant (*p* < 0.05) increase in the volume of GC-dough could be observed with the increase in proofing time. After 40 min proofing, the volume of GC-dough was smaller than those of the blank. The results might indicate that the addition of the present GC could reduce the volume of dough within 20–60 min of proofing. For the MCC-dough, the significant (*p* < 0.05) increase in volume could be observed between 20 and 40 min. A rigid dough enhanced by the GC was indicated as highly resistant to deformation, which limits the expansion of gas cells in the GC-dough during proofing [12]. Moreover, the effect of GC on the proofing behavior of GC-dough might be also assigned to the strengthening effects on the texture of present bread.

The effect of GC on the pH of GC-dough is shown in Table 2. Although a reduction in pH value could be observed in GC-dough, there was no significant (*p* < 0.05) difference in dough either with or without GC and MCC in 20–60 min of proofing. Such results might be attributed to the esters, which could neutralize the lactic acid in dough proofing, and buffering the pH in proofed GC-dough and MCC-dough. Similar results were supported by dough characteristics and bread quality with psyllium husk fiber [32]. In that study, the values of pH indicate that the crumb with fiber recorded a significantly lower pH value. Moreover, the buffering capacity of the GC and MCC might be related to the high cation exchange capacity shown in Figure 2A.

The color quality of bread could greatly affect its acceptance by consumers. In this study, no significant (*p* < 0.05) effect of the GC and MCC on the color of dough could be observed, as shown in Table 2. The light values of GC-dough were higher than those of blank control and MCC. Furthermore, the GC-dough proofed for 40 min showed the highest light values of 321.39 ± 17.09. The results suggested that the addition of the GC might improve the color of the dough, which might indicate the potential application of the GC in commercial production.

### 3.9. Effect of the GC on Gluten of Bread

The gluten, which consists of glutenin and gliadin, evidently affects the structure and texture of the fermented dough. We evaluated the effects of the GC on the content and water holding capacity (Figure 3). An obvious reduction could be observed in the gluten content of GC-dough (24.96 ± 1.22%) as compared to blank control (31.50 ± 0.89%). Besides the reduced content of high-gluten flour, the high water holding capacity of GC (Table 1) might also contribute to such results, and inhibit the formation of gluten in dough [33].

Moreover, such a high water holding capacity of GC was further supported by a similar high water holding capacity of the GC-gluten [34]. As shown in Figure 3, the water holding capacity of the GC-gluten was 1.71 ± 0.08 g/g, which was significantly (*p* < 0.05) higher than that of the blank control. The results were related to high water holding capacity of the GC, as shown in Table 1. In previous studies, cellulose was reported to affect protein water availability, and delay the development of gluten [9]. Therefore, the high water holding capacity of the GC-gluten would suggest that the gluten network was differently developed [9].

### 3.10. Structure of the Protein in GC-Gluten

In addition, the water holding capacity of GC-gluten has a critical influence on the secondary structure of gluten protein. The secondary structure of protein in GC-gluten was further studied, and the results are shown in Table 3. Firstly, the content of glutenin in GC-gluten was significantly (*p* < 0.05) more than those of blank control, while a significant (*p* < 0.05) reduction in gliadin in GC-gluten could also be observed. Furthermore, both the content of glutenin and gliadin in the GC-gluten was significantly (*p* < 0.05) more than those gluten of the MCC. Carlos et al. [25] indicated that the addition of cellulose allowed the formation of a resistant gluten network, which lies in a large number of interactions between glutenin and gliadin.

Then, the contents of α-helix, β-sheet and random of the GC-gluten significantly (*p* < 0.05) increased as compared with blank control (Table 3), while those of β-turns were significantly (*p* < 0.05) reduced. The high holding capacity of GC-gluten might be attributed to such results. Previous studies had suggested that water molecules could be held between protein chains, weakening the hydrogen bonding between glutenin and gliadin. The elasticity of the gluten network might decrease with viscous behavior in a Maxwell representation for dough relaxation [35].

### 3.11. Effect of the GC on Starch of Bread

For the further evaluation of the effect of GC in the structure of the present bread, the solubility and swelling of starch were determined. As shown in Table 4, the solubility and swelling of starch were significantly (*p* < 0.05) lower than the dough without GC. Correa et al. [36] proposed that hydrocolloids, which are formed by cellulose, could modify the pasting profiles of wheat flour by means of hindering the hydration of starch granules or the lixiviation of amylose. Therefore, the hydrocolloids formed by the GC in GC-dough might be attributed to such poor solubility and swelling of the starch.

### 3.12. Characterization of Bread with Added GC

The morphology and texture of the present bread with added GC were determined in this study. The images of SEM investigated the structure of GC-bread (Figure 4). A cross-linked network formed by gluten protein could be observed in the present bread. The intermolecular covalent cross-linking by the disulfide bond of GC-gluten protein might be attributed to such morphology. Moreover, the concentration-dependent destruction of the network could be obviously observed with the addition of the GC in the present bread, and the granules of GC might have hindered the proper arrangement of the gluten proteins [37]. With the increasing ratio of the GC from 1% to 7%, the smooth surface of the cross-linked network was developed. The GC-bread demonstrated the presence of a network mostly composed of gluten films, showing a more closed and less cross-linked appearance, but also a more orientated network. Such microstructure of GC-bread was related to previous studies [37], which suggested that dough prepared with hydroxypropyl methylcellulose presented an open and cross-linked gluten network. On the contrary, different crumb structures were reported by the study of Ramon et al., in which more irregular crumb and larger cell size could be observed in the bread with added psyllium husk fiber [10].

The TPA further confirmed the effect of GC on the texture of the present bread (Figure 5). With the addition of the GC, a significant effect (*p* < 0.05) could be observed on the hardness, adhesiveness and chewiness as compared with the blank control. Moreover, the hardness of the present bread significantly (*p* < 0.05) increased with the increasing content of the GC. Ren et al. [13] had reported similar results in the study of gluten-free bread enhanced by cellulose. That study speculated that both flour and cellulose particles were rigid fillers. The cellulose particles are more rigid than the hydrated flour particles in the bread, competing for water and further increasing the rigidity of the dough. Similar high hardness was also supported by the bread formulated with blackcurrant dietary fiber [8]. Therefore, the bread is strengthened by the addition of cellulose [7]. Furthermore, the bread with added 5% of GC showed the highest adhesiveness of the present bread, while the highest chewiness could be observed in the bread with added 7% of GC. The GC enhanced bread showed a desirable textual property in the manufacture [3], where it is also beneficial to the handling properties for storage. Such results might be supported by the cross-linking morphology shown in Figure 4, suggesting the enhanced quality of the bread induced by improved characterization of GC-dough and GC-gluten.

The improved texture was further confirmed by the sensory evaluation of the bread added to GC (Figure 6). In comparison to the bread without GC, significantly (*p* < 0.05) increased values of aroma, tactility, state, shape, texture and acceptability could be observed in the bread with added 1% of GC. Furthermore, the concentration-dependent reduction in the value of tractility and shape was shown with the addition of GC (1%~7%). Wheat bread is expected to have a softer and more elastic texture, and provide a comfortable taste [38]. Therefore, the increasing hardness with the addition of GC might be related to the results, which resulted in the poor sensory characterization. The results indicated that the bread with added 1% of GC provided the highest sensory characterization, suggesting the best strategy for further study.

### 3.13. Cold-Stored Stability of the GC-Bread

For the further evaluation of the stability of the present bread, the core moisture and hardness were studied in 168 h cold storage at 4 °C. As shown in Figure 6A, a significant effect of the GC on the core moisture of the present bread could be observed in the 168 h cold storage. In the 72 h storage, the core moisture of bread with added 1%~5% of GC was higher than that without GC. In addition, with the increasing storage time from 120 to 168 h, the bread with added 5% and 1% of GC showed the highest value of core moisture, respectively. The high water holding capacity of the GC might attributed to such high core moisture in the present bread, which was in agreement with the results in Table 1 and Figure 3. In addition, the denser crumb structure might also be related to such high core moisture, where water evaporation within the gas cells was restrained, which was supported by the studies by Nie et al. [39], which reported that the cellulose-enhanced bread increased the freezable water of the system, improving the molecular mobility of water molecules.

As shown in Figure 7, the hardness of the bread significantly (*p* < 0.05) increased with the increasing storage time to 168 h. Moreover, after 120~168 h cold storage, the hardness of the bread with added 1%~5% of GC was significantly (*p* < 0.05) lower than that of bread without GC. Furthermore, the bread with added 1% of GC showed the lowest hardness after 168 h cold storage, which supported the TPA and best sensory characterization, as shown in Figure 5 and Figure 6.

As shown in Table 5, the hardness of the present bread in 168 h cold storage was further studied by the kinetic Avrami formula. As compared to the bread without GC, low Avrami index and K values could be observed in the bread with added 1%~7% GC, increasing with the addition of GC. Furthermore, the bread with added 1% of GC showed the lowest Avrami index and K values after 168 h cold storage, suggesting the best cold-stored stability in the present bread. The results were in agreement with the TPA and sensory characterization, further confirming the significant effect of the GC on the stored stability of the present bread.

In the present work, the bread with added 1% of GC provided the highest sensory characterization and best cold-stored stability, which suggested the best strategy for further study. The results of GC made them a good candidate as effective bread improver. Even when the interaction between the characterization of the GC and the texture of the GC-bread has not been completely understood, it should be further addressed.

This study has the following limitations. First, ethical restrictions prevented the use of MCC in human sensory evaluation experiments, which may have constrained direct comparative analyses with GC. Second, the relatively limited sample size could affect the generalizability of the findings. Future studies should prioritize expanding the sample size and investigating functional properties of alternative cellulose sources, particularly those derived from marine biomass, to further validate and broaden the applicability of these results in bakery product development.

## 4. Conclusions

In conclusion, this study demonstrates that Gagome kelp cellulose (GC) exhibits superior functional properties, including enhanced water and oil holding capacity, cholesterol adsorption, and glucose diffusion inhibition, compared to microcrystalline cellulose (MCC). GC also significant influences on the properties of dough, gluten, starch, and bread, improving texture, glutenin content, and cold storage stability. Notably, bread containing 1% (m/m) GC showed optimal sensory characteristics and storage performance, highlighting its potential as a natural bread improver. These findings suggest that GC is a promising biocompatible ingredient for developing functional bread formulations, offering a sustainable strategy for enhancing bakery products. Future studies should focus on scaling up production and evaluating consumer acceptance to further validate its commercial applicability.

## Figures and Tables

**Figure 1 foods-14-01246-f001:**
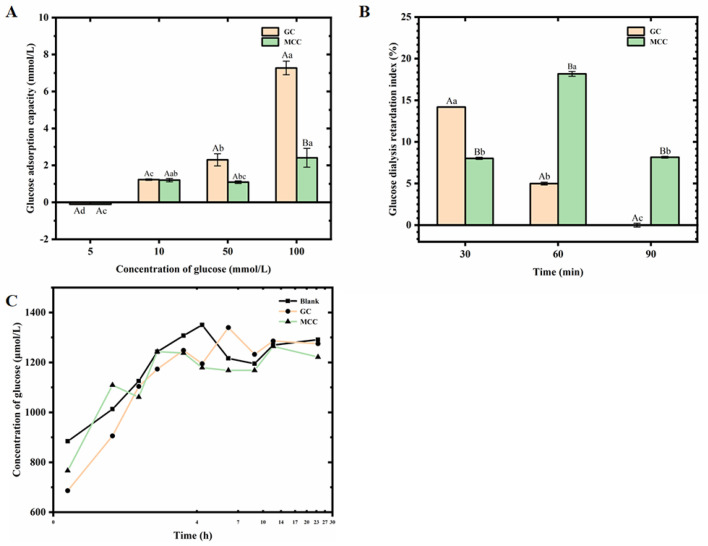
Glucose adsorption capacity (**A**), GDRI value (**B**) and inhibition of glucose diffusion capacity (**C**) of the GC. The deionized water was used as the blank control. The capital letters in the table indicate a significant difference (*p* < 0.05) with treatment, while the lowercase letters indicate a significant difference (*p* < 0.05) with concentration of glucose (**A**) and time (**B**).

**Figure 2 foods-14-01246-f002:**
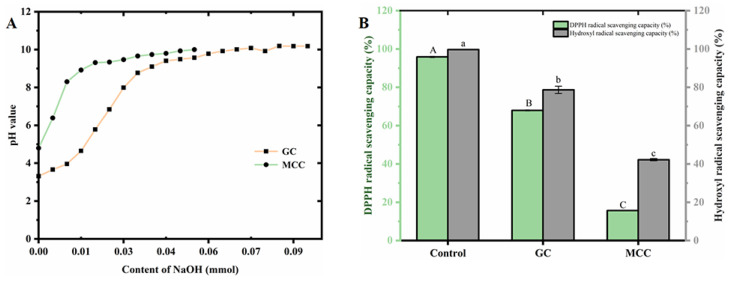
Cation exchange capacity and antioxidant activity of the GC. (**A**) Cation exchange capacity of the GC; (**B**) DPPH free radical and hydroxyl radical scavenging rate (**B**) of the GC. L-ascorbic acid was used as the control. The letters in the figures indicate a significant difference (*p* < 0.05, *n* = 6).

**Figure 3 foods-14-01246-f003:**
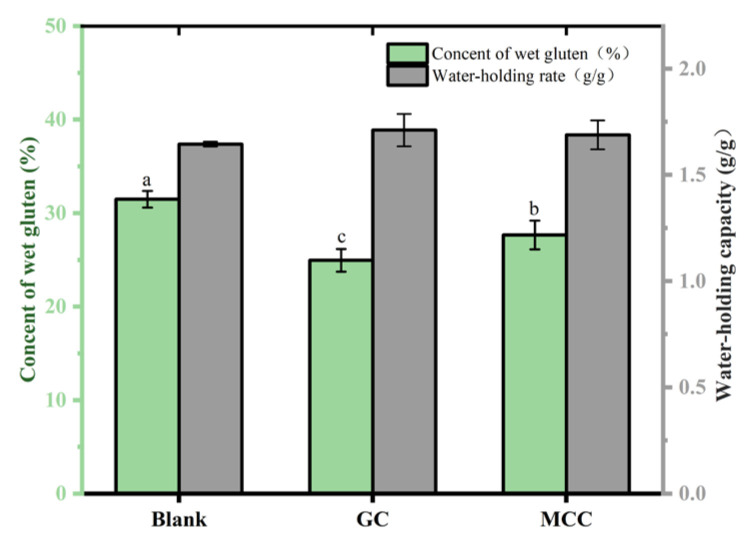
Wet gluten content and water holding capacity of the GC-gluten. The gluten without cellulose was used as a blank control. The letters in the figures indicate a significant difference (*p* < 0.05).

**Figure 4 foods-14-01246-f004:**
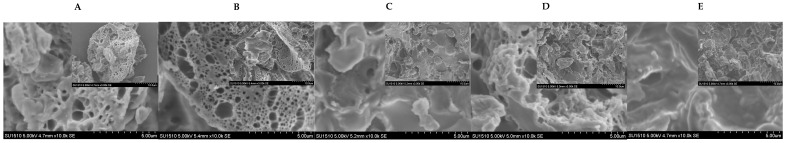
SEM images of the GC-bread. (**A**) bread without GC; (**B**) bread with added 1% of cellulose; (**C**) bread with added 3% of cellulose; (**D**) bread added with 5% of cellulose; (**E**) bread with added 7% of cellulose. The observations were performed at an accelerating voltage of 5.00 kV. The scale bar of the big figure is 5 μm (10.0 K), while that of the small is 10 μm (3.00 K).

**Figure 5 foods-14-01246-f005:**
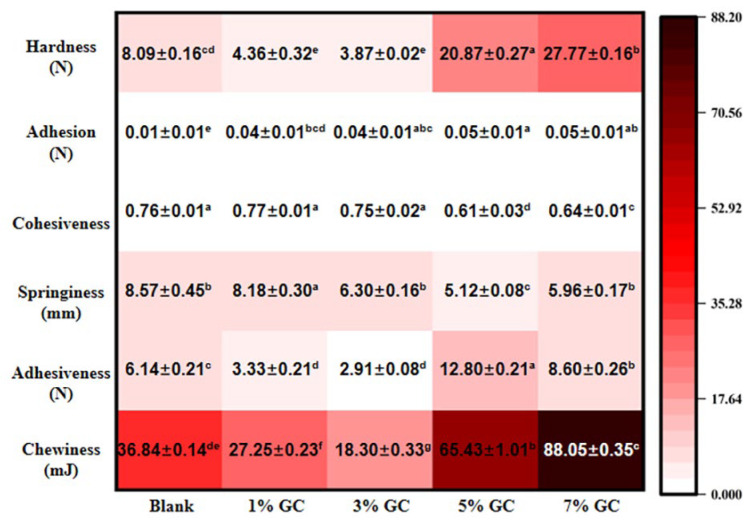
TPA of the GC-bread. Results were expressed as mean ± std., where std. is the standard deviation, *n* = 10. The lowercase letters in the table indicate a significant difference (*p* < 0.05).

**Figure 6 foods-14-01246-f006:**
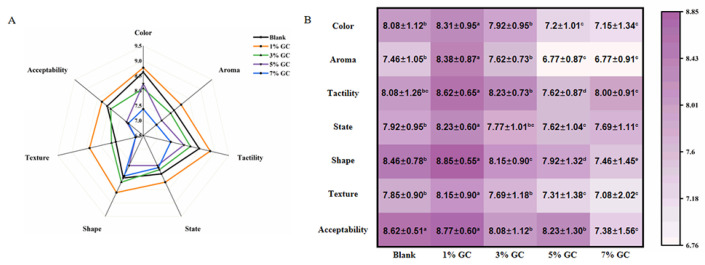
Sensory evaluation of the GC-bread. (**A**) radar map; (**B**) heatmap. Results were expressed as mean ± std., where std. is the standard deviation, *n* = 3. The lowercase letters in the table indicate a significant difference (*p* < 0.05).

**Figure 7 foods-14-01246-f007:**
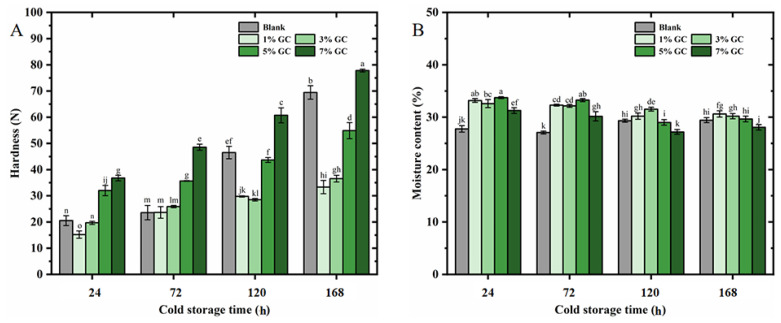
Cold storage stability of the GC-bread. (**A**) The core moisture of cold storage GC-bread; (**B**) the hardness of cold storage GC-bread. The lowercase letters in the table indicate a significant difference (*p* < 0.05).

**Table 1 foods-14-01246-t001:** Characterizations of the GC and MCC ^#^.

Characterizations		MCC	GC
Water holding capacity (g/g)		2.98 ± 0.21	3.41 ± 0.10 *
Oil holding capacity (g/g)		3.75 ± 0.21	4.11 ± 0.06 *
Unsaturated fat adsorptive capacity (mg/g)		1.30 ± 0.04	1.35 ± 0.04
Saturated fat adsorptive capacity (mg/g)		1.62 ± 0.17	1.73 ± 0.20 *
Cholesterol-absorbing capacity (mg/g)	pH 2.0	5.88 ± 0.42	4.48 ± 0.14 *
	pH 7.0	7.60 ± 0.37	5.47 ± 0.38 *
α-Amylase inhibitory activities (%)		24.71 ± 0.20	14.31 ± 0.21 *

^#^ Results are expressed as mean ± std., where std. is the standard deviation, *n* = 10. The mark “*” in the table indicates a significant difference (*p* < 0.05) between the MCC and GC.

**Table 2 foods-14-01246-t002:** The volume, pH value and color of the GC-dough ^#^.

	Sample	20 min	40 min	60 min
volume	Blank	24.44 ± 0.58 ^Ac^	31.50 ± 0.50 ^Bb^	34.70 ± 0.46 ^Aa^
	GC	23.33 ± 0.58 ^Ac^	26.93 ± 1.01 ^Cb^	32.07 ± 1.01 ^Ba^
	MCC	23.83 ± 0.50 ^Ab^	33.17 ± 0.29 ^Aa^	33.34 ± 1.04 ^ABa^
pH value	Blank	5.18 ± 0.08 ^Aa^	5.08 ± 0.10 ^Aa^	5.10 ± 0.06 ^Aa^
	GC	5.13 ± 0.10 ^Aa^	5.01 ± 0.02 ^Aa^	5.01 ± 0.13 ^Aa^
	MCC	5.16 ± 0.03 ^Aa^	5.11 ± 0.11 ^Aab^	5.05 ± 0.05 ^Ab^
color	Blank	304.41 ± 3.45 ^Aa^	294.02 ± 0.92 ^Aa^	311.62 ± 1.74 ^Aa^
	GC	299.83 ± 2.73 ^Aa^	321.39 ± 1.80 ^Aa^	316.14 ± 1.02 ^Aa^
	MCC	303.56 ± 3.25 ^Aa^	295.15 ± 4.08 ^Aa^	306.14 + 1.02 ^Aa^

^#^ Results were expressed as mean ± std., where std. is the standard deviation, *n* = 3. The capital letters in the table indicate a significant difference (*p* < 0.05) with treatment, while the lowercase letters indicate a significant difference (*p* < 0.05) with time.

**Table 3 foods-14-01246-t003:** Composition of the GC-gluten and the secondary structure of the gluten-protein ^#^.

Content (%)	Blank	GC	MCC
Glutenin	66.54 ± 0.02 ^c^	71.76 ± 0.05 ^a^	69.05 ± 0.06 ^b^
Gliadin	36.56 ± 0.01 ^a^	30.62 ± 0.01 ^b^	29.33 ± 0.01 ^c^
Gliadin/Glutenin	54.96 ± 0.01 ^a^	42.81 ± 0.04 ^b^	42.70 ± 0.05 ^b^
α-helix	15.31 ± 6.23 ^c^	18.51 ± 4.82 ^b^	20.35 ± 6.28 ^a^
β-sheet	36.80 ± 6.23 ^c^	40.92 ± 1.69 ^b^	47.16 ± 6.88 ^a^
β-turn	32.15 ± 7.50 ^a^	24.65 ± 2.07 ^b^	24.65 ± 2.07 ^b^
Random coil	15.58 ± 0.21 ^b^	18.98 ± 2.02 ^a^	15.68 ± 0.01 ^b^

^#^ Results were expressed as mean ± std., where std. is the standard deviation, *n* = 3. The lowercase letters in the table indicate a significant difference (*p* < 0.05).

**Table 4 foods-14-01246-t004:** Effect of the GC on the solubility and swelling capacity of starch ^#^.

Sample	Solubility (%)	Swelling Capacity (%)
Blank	33.91 ± 0.63 ^a^	13.95 ± 0.09 ^a^
GC	23.83 ± 0.36 ^c^	12.73 ± 0.05 ^b^
MCC	28.45 ± 0.48 ^b^	14.18 ± 0.44 ^a^

^#^ Results were expressed as mean ± std., where std. is the standard deviation, *n* = 3. The lowercase letters in the table indicate a significant difference (*p* < 0.05).

**Table 5 foods-14-01246-t005:** Kinetic parameters of hardness changes on the GC-bread ^#^.

Content of the GC	F_0_	F_7_	Avrami Index (*n*)	K (d^−1^)	R^2^
0	4.67 ± 0.74 ^c^	69.47 ± 2.56 ^b^	1.37 ± 0.02 ^d^	8.62 ± 0.56 ^a^	0.95 ± 0.02 ^d^
1% GC	3.87 ± 0.02 ^e^	33.33 ± 2.54 ^e^	1.08 ± 0.03 ^c^	3.95 ± 1.45 ^d^	0.91 ± 0.01 ^c^
3% GC	4.36 ± 0.31 ^d^	36.57 ± 1.27 ^d^	1.06 ± 0.02 ^e^	3.90 ± 0.55 ^d^	0.86 ± 0.01 ^e^
5% GC	13.43 ± 0.20 ^b^	54.90 ± 3.08 ^c^	1.16 ± 0.02 ^b^	5.08 ± 0.48 ^c^	0.90 ± 0.02 ^d^
7% GC	23.13 ± 2.46 ^a^	77.83 ± 0.55 ^a^	1.31 ± 0.03 ^a^	7.33 ± 1.88 ^b^	0.99 ± 0.01 ^a^

^#^ Results were expressed as mean ± std., where std. is the standard deviation, *n* = 3. The lowercase letters in the table indicate a significant difference (*p* < 0.05) with treatment.

## Data Availability

The data presented in this study are available on request from the corresponding author. The data are not publicly available due to privacy restrictions.

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
