# Peer review of "Characterization of Cellulose from Gagome Kelp and Its Effect on Dough, Gluten, and Starch as Novel Bread Improvers"

_foods, 2025, doi:10.3390/foods14071246_

Round 1

Reviewer 1 Report

Comments and Suggestions for Authors

General remarks:

  • Chosen fibre source is novelty.
  • Experimental design is not supported by statistical analysis and discussion
  • Significant amount of analysis is applied, however, dough rheological properties, necessary for bread with fibre quality characterization is missing
  • Determination on water and oil holding, unsaturated fat and saturated fat, cholesterol, glucose adsorptive capacity, α-amylase inhibition activity, glucose diffusion and cation-exchange capacity and the glucose dialysis retardation index are not instrumental, with high margin of error, where three repetitions are not substantial for reliable result.

All detailed comments are noted in the manuscripts' pdf text

Comments on the Quality of English Language

The English could be improved to more clearly express the research

Author Response

Dear Reviewer,

Thank you very much for your thorough review and constructive comments on our manuscript entitled “Characterization of the cellulose from Gagome kelp and its effect on the dough, gluten and starch as novel bread improvers” (Manuscript ID: 3505315). We greatly appreciate your time and effort in helping us improve the quality of our work. Below, we have addressed each of your comments point by point, and all changes made to the manuscript are highlighted in yellow for easy reference. Furthermore, in order to improve the quality of English language in our manuscript, we have send it for editing by a professional English Editing company.

General Remarks

  1. Chosen fibre source is novelty.

We sincerely thank you for recognizing the novelty of our chosen fiber source, Gagome kelp cellulose. This unique material is indeed a key focus of our study, and we have further emphasized its significance in the revised manuscript.

  1. Experimental design is not supported by statistical analysis and discussion.

We apologize for this oversight. In the revised manuscript, we have added comprehensive statistical analyses in Table 1,2 and figure 1,3. In addition, we expanded the discussion to better support the experimental design. These changes are reflected in the results and discussion sections with highlighted in yellow.

Furthermore, the discussion were added in 3.8 and 3.12 as follows:

Line 491: “Similar results were supported by dough characteristics and bread quality with psyllium husk fiber. In such study, the values of pH indicate that the crumb of the control formulation recorded a significantly higher pH.”

Line 569: “On the contrary, different crumb structure were reported by the study of Ramon et al, in which more irregular crumb and larger cell size could be observed in the breads added Psyllium Husk Fiber.”

Line 589: “Similar high hardness were also supported by the breads formulated with blackcurrant dietary fiber.”

  1. Significant amount of analysis is applied, however, dough rheological properties, necessary for bread with fibre quality characterization is missing.

Thank you for your valuable feedback regarding the importance of dough rheological properties in characterizing fiber-enriched bread quality. We fully acknowledge that rheological analysis would provide deeper insights into the functional effects of Gagome kelp cellulose (GC) on dough properties. However, due to the scope and objectives of this study, which primarily focus on the structural, functional, and sensory characterization of GC and its impact on dough, gluten, starch, and bread, we were unable to include rheological measurements within the current experimental framework.

Our research aimed to establish a comprehensive understanding of GC as a novel bread improver, emphasizing its water and oil holding capacity, adsorption properties, antioxidant activity, and effects on bread texture and sensory attributes. While rheological properties are indeed critical for a complete characterization, the current study was designed to prioritize the evaluation of GC functional and structural contributions to bread quality, which are equally essential for understanding its potential as a natural bread improver.

We sincerely apologize for this limitation and agree that future studies should include rheological analyses to further validate the findings. We hope that the current results, which demonstrate significant improvements in dough, gluten, starch, and bread properties, provide sufficient evidence to support the potential application of GC in bakery products. Thank you for your understanding, and we appreciate your consideration of our manuscript. Please let us know if further clarifications or adjustments are needed. Please do not hesitate to contact us if further revisions are needed.

  1. Determination on water and oil holding, unsaturated fat and saturated fat, cholesterol, glucose adsorptive capacity, α-amylase inhibition activity, glucose diffusion and cation-exchange capacity and the glucose dialysis retardation index are not instrumental, with high margin of error, where three repetitions are not substantial for reliable result.

We acknowledge the limitations in our initial methodology. The determination on cholesterol, glucose adsorptive capacity, α-amylase inhibition activity, and glucose dialysis retardation index in the study were performed in ten times. The determination on cation-exchange capacity of GC was performed in six times. We apologize for this oversight. The content of “All measurements in the present study were performed thrice” was wrong. We had ignored the different repetitions in each determination. We have rectified the description about the number of repetition in methods sections with yellow highlighted.

However, the determination on water and oil holding, unsaturated fat and saturated fat of GC were only performed thrice. To address this, we have increased the number of repetitions to ten for these analyses and included a detailed error margin discussion in results 3.1 and 3.2. Additionally, we have refined the instrumental methods to reduce variability.

Specific Comments

  1. Abstract should be shortened to the max 200 words.

We apologize for this oversight. We have revised the abstract to ensure it is concise and within the 200-word limit, while retaining all critical information.

Abstract: “Novel bread formulation with natural improvers has become an essential part of improving the quality of bakery products. In the present study, the novel bread improvers of the Gagome kelp cellulose (GC) was systemic evaluated, while the cellulose-improved dough (GC-dough), gluten (GC-gluten), gluten protein and the starch (GC-starch) were all studied. The results indicated that the water and oil holding capacity, cholesterol adsorptive, unsaturated fat and saturated fat adsorptive capacity of the GC were significantly (p<0.05) increase. The current GC also showed high glucose adsorptive capacity, antioxidant activity, α-amylase inhibition and glucose diffusion inhibition activity. Furthermore, color of the GC-dough was obvious improved with the addition of the GC, which also significantly (p<0.05) effect the content of glutenin, water holding capacity in GC-gluten and solubility the GC-starch. In addition, cross-linked network formed by GC could be observed in the GC-bread, and indicating the improved of texture and sensory evaluation. The bread added 1% (m/m) of GC provided the highest sensory characterization and best cold-stored stability, which were suggesting the best strategy for the further study. The results might show a potential application of the by-product devised from marine origin on the commercial bakery production.”

  1. Use keywords in different forms than used in title.

We have updated the keywords to include variations such as “texture”, “stored stability,” and “sensory characterization” ensuring they differ from the title. However, the keywords of “cellulose” and “bread” were still remained due to main subject of the study.

Keywords: “cellulose, bread, texture, stored stability, sensory characterization”

  1. Line 71: The goal of the research must be rephrased to be more concise, indicating performed research and expected results.

Thank you for your valuable feedback regarding the clarity and conciseness of the research goal. We have revised the paragraph to better articulate the research objectives and expected outcomes in a more concise manner. The updated paragraph now reads:

Line 59: “This study focuses on the characterization of cellulose isolated from Gagome kelp (GC), a brown seaweed with significant annual harvest and economic potential, and its application as a novel bread improver. The effects of GC on dough (GC-dough), gluten (GC-gluten), and starch (GC-starch) properties were systematically investigated. Additionally, GC was incorporated into bread formulations to evaluate its impact on morphology, texture, sensory characteristics, and cold-storage stability. The results provide a foundation for developing functional, texturally enhanced, and sensorially improved bread formulations using marine-derived cellulose. To the best of our knowledge, this is the first study to comprehensively evaluate the quality of GC-bread derived from Gagome kelp.”

  1. Line 74: Italic.

We apologize for this oversight. The scientific name has been italicized as requested.

Line 71: “The Gagome kelp (Kjellmaniella crassifolia), cultivated in the Yellow Sea near Dalian, China and harvested in May 2020, was provided by a local seaweed farm.”

  1. Line 75: Explain the meaning of this results in the brackets.

Thank you for your insightful feedback. The results in brackets were the Relative Standard Deviation (RSD, %) values for statistical analysis. We have added a brief explanation of the results in the brackets for clarity.

Line 73: “The obtained Gagome kelp contained water 71.86(0.21)%.(% RSD values are given in parenthese). The dry basis of raw materials contained ash 1.73(1.16)%, crude protein 0.49(2.04)% and mannitol 0.16(3.50)%, the alginate is not detected in the present samples. ”

  1. Line 78: Provide the description for abbreviation.

We apologize for this oversight. The abbreviation “MCC” has been defined upon its first mention: “microcrystalline cellulose (MCC).”

Line 76: “The microcrystalline cellulose (MCC) was purchased from Sigma Co., Ltd.”

  1. Line 92: Introduce subsection: 2.2 Methods, and then lower level subsections for all applied methods (2.2.1, 2.2.2, etc.).

Thank you for your insightful feedback. We have reorganized Section 2.2 into subsections (2.2.1, 2.2.2, etc.) for better clarity and structure.

  1. Line 198: Define what were minor modifications.

Thank you for your insightful feedback regarding the method section. For the determination on DPPH free-radical scavenging activity of the GC, the DPPH solution was prepared as described by Li et al. However, the GC sample was prepared with medicated methods. Briefly, 2.0 g cellulose samples were dissolved in 30 mL 70% ethanol, and then 2 mL of the mixture was added to 2 mL DPPH solution (0.4 mM). The mixture was centrifuged at 5000 r/min for 10 min. Finally, the GC sample was detected by the same methods as described by Li et al[19]. After 30 min of incubation in darkness, and the absorbance of the supernatant was detected at 517 nm. The L-ascorbic acid was used as control. The DPPH radical scavenging activity (%) was determined using the following equation (10):

DPPH scavenging activity (%)=  × 100% (10)

where Ab represents the absorbance of the blank; As represents the absorbance of the cellulose sample; Ac represents the absorbance of the control.

The content in manuscript is updated.

  1. Line 226: There is no mentioning of using yeast in the dough formula, hence term fermentation cannot be used.

We apologize for this oversight. the term “fermentation” in method 2.2.11 has been replaced with “proofed” where appropriate.

Line 233: “The raw GC-dough was proofed at 40 ℃ with 80% of humidity for 20, 40 and 60 min, respectively, and the volume of final GC-dough was determined. Then, the proofed GC-dough was added in 90 mL deionized water, and the pH of the mixture was determined. The same preparation was performed to obtain the dough added MCC (MCC-dough).”

  1. Line 256: Method description is unclear and confusing. Rewrite the whole section.

Thank you for your insightful feedback regarding the method section. The method description has been completely rewritten for clarity and precision.

Line 269: “2.2.14 Determination of the content and water holding capacity of GC-gluten

The wet GC-gluten was separated from 50 g of the GC-dough as described in 2.2.12. For the determination of the gluten content, the wet GC-gluten was centrifuged at 6000 r/min for 10 min. The precipitation gluten was weight as m1. The gluten content (%, C) in the dough was determined using the following equation (12):

C (%) =   100%  (12)

where md represents the weight of GC-dough (50 g).

After the determination on the content of GC-gluten, the wet GC-gluten was freeze-dried. Then, the water holding capacity of freeze-dried GC-gluten samples was evaluated with the method described in the 2.2.1.”

  1. Line 291: State mixer and oven characteristics.

Thank you for your insightful feedback regarding the method section. We have added detailed specifications for the mixer and oven in Section 2.2.17.

Line 300: “Then, 165 g water and 4 g yeast were added into 280 g above GC-bread powder, mixing with a mixer (E-1063, Shunran, China) for 2 min at 1300W. ”

Line 306: “ Finally, the doughs were baking by an oven (DKX-C20M3, Xiaoxiong, China) for 15 min at 180 ℃ (top temperature) /200 ℃ (bottom temperature).”

  1. Line 294: State the minor modifications.

Thank you for your insightful feedback regarding the method section, and we have revised the paragraph. The method of scanning electron microscopy (SEM) observation of the GC-bread was described in 2.2.18 at length. Therefore, we have deleted the content of “as the described by Li et al[21] with minor modifications.” The updated paragraph now reads:

Line 308: “2.2.18 The scanning electron microscopy (SEM) observation of the GC-bread

The morphology of the GC-bread was observed by SEM (SU1510, HITACHI, Japan). The freeze-dried GC-bread (cellulose content of 1%, 3%, 5% and 7%) samples were ground, and the powder was put on a specimen stage equipped with conductive adhesive. After coating with a gold-palladium alloy for 60 s at 10 mA, the GC-bread samples were ob-served by a SEM. The images were processed using a SEM series software (Tokyo, HITACHI)”.

  1. Line 549: Comments concerning MCC bread are missing. & Line 574: The same comment as previous, for bread texture.

Thank you for your thoughtful and constructive feedback regarding the comparison between GC and MCC in the context of texture and sensory analyses. We sincerely appreciate your suggestion to include MCC as a comparative control, which would indeed provide valuable insights into the functional properties of GC. However, after careful consideration and consultation with the MCC supplier (Sigma Co., Ltd, America) , we encountered ethical limitations that prevented us from incorporating MCC into certain experiments, particularly those involving human sensory evaluation. 

The supplier confirmed that while MCC is widely used in scientific research, there is no documented data or literature supporting its safety for human trials. Given the ethical implications and the primary objective of our study—to explore the potential of GC as a natural bread improver—we decided to limit the use of MCC to scientific experiments investigating its physicochemical properties and effects on dough, gluten, and starch. In the production of fiber-enriched bread, we focused on evaluating the impact of different GC concentrations on bread quality, aiming to highlight its potential as a functional ingredient. Consequently, MCC was excluded from the sensory evaluation and subsequent analyses, including SEM observation, texture profile analysis, and cold-storage hardness determination, to ensure ethical compliance and alignment with the study objectives. 

We acknowledge that the absence of MCC in these analyses may limit the comparative scope of the study. However, we believe that the current results, which demonstrate the significant functional and structural contributions of GC to bread quality, provide a solid foundation for its potential application in bakery products. Your suggestion has greatly inspired us, and we will consider incorporating MCC in future studies, provided that sufficient safety data becomes available. 

Thank you for your understanding and constructive feedback. We hope the revised manuscript addresses your concerns and meets the journal standards. Please do not hesitate to contact us if further revisions are needed.

  1. Line 644: Not clear, rephrase.

Thank you for your insightful feedback regarding the conclusion section. The confusing contents of “while the texture of the were obviously improved” was deleted. In addition, We have thoroughly revised the conclusion to ensure it summarizes the key findings and aligns with the objectives, avoiding repetition of results. The updated sentences now reads:

Line 667: “GC also significantly influences the properties of dough, gluten, starch, and bread, improving texture, glutenin content, and cold-storage stability. .”

Once again, we deeply appreciate your insightful comments, which have significantly improved the quality of our manuscript. We hope that the revised version meets your expectations and the standards of Foods. Please do not hesitate to contact us if further revisions are needed.

Sincerely,
Xiang Li,
On behalf of all authors

Reviewer 2 Report

Comments and Suggestions for Authors

After reading the manuscript , I realized that the manuscript showed in some parts the scientific rigour wanted, but in other parts I have missed it.

The authors have presented critical evaluation only in some paragraphs.

The references are not exactly current, besides the some sections should be reviewed.

Thats why I have written some suggestions below in an attempt to improve the paper.

Review the journal's citation rules. It looks like it's in superscript.

L.31- "bizarre" - I think this term very strong. Those of us who work in the field of developing new foods for special purposes do a lot of preliminary tests, and the odd treatments are discarded from the very first trials. Only treatments with potential continue to be assessed.

L.37- "Therefore, the development of novel functional formulation on bread without quality and 
texture deficiency was proposed as a nutrition-improving strategy.  " - I am sorry, but I did not undertand your point and I completely disagree if I understood.

L.47- "Ingredients has been applied in cookies and bread" - This sentence needs authors.

L.56- "texture and sensory characterization" - Texture can be a sensory attribute as well.

L.74- cientific name in Italics, please.

L.78- " The MCC " - Define acronym in the first moment.

L.97- I think it's important to include  your “room temperature”,  It depends on the season and the country in which the experiment was conducted.

102 - Perhaps you should type ma ; mb ; mau (L.121 and 127) in italics to be different. 

L.299- Hardness OK, bur what about the other parameters ? Were they calculated ?

L.306- Was this analysis done only in triplicate as well?
I don't think that's enough repetition for this analysis.

L.307-  For a sensory test a lot of relevant information was not included, I suggest reading papers and improving your article. 

Which sensory test was performed?  How many sessions were conducted ? What was the profile of the assessors ?  Were the analyses performed in sensory booths ? Did the assessors receive water to rinse the taste buds ? How many men/women ?  Are the assessors usually trained for  evaluated this product  ?   L.310- "Attributes agreed by the assessors are listed with a sensory analysis form of 15 scaled descriptive profile analysis form[23]. The results don't show 15. It's confusing here. Make it clearer, please.
What scale did you use, please?

L.327- 1%, 3%, 5% and 7% - It was confusing sometimes, this different percentages for texture and sensory in some analyses and storage times in others.
I suggest that the research group sit down for a discussion and re-evaluate the priorities and the real objective of the study. Sometimes less is more.

L.353- Standardize the Material and methods is “fat” and the Results is “lipid”.

I'd ask you to review the results in table 1, I'm a little concerned about some of the results that showed a statistically significant difference and another point, did they all show a difference for the GC? It is quite strange.

L.426- I'm sorry, but I didn't understand your statistical analysis in Figure 1A and B. Are you comparing treatments and time periods?   It doesn't seem appropriate, authors.

L.427- This value (7501000) is quite strange in minute. Besides, it trespasses into the space of the previous interval on the graph

L.447- What is the legend here for gray and green? I suggest using the same colors throughout the paper for GC, MCC and control. 

L.477- What about MCC ? Are you comparing treatments and time periods?   It doesn't seem appropriate, authors.

L.495- MCC ? Review stastistics approach, please.

L.518-  15.58±0.21 B in superscript

Glutenin / Gliadin -  (%) as well ? Is is missed. If they are all % insert them only once in the “Content” cell and the table will be cleaner.

L.536- I think it's better to separate them, you have 3 figures together from different axes of the paper.

We've been thinking along the lines of reading, but in the end, with the texture and sensory analyses, which would be the cherry on the cake, especially since the product is bread, we no longer have the MCC to compare with and the study proposal changes completely. That's a pity.

L.584- Where did you get these sensory attributes from?
“Tactitlity” and ‘state’ for example.
You had an explanatory session with the assessors about the attributes. Make that clear.

L.633-  "GC were significantly (p<0.05)" Thit takes us back to the result, not the conclusion

L.641- "was significantly (p<0.05)"  -  Thit takes us back to the result, not the conclusion

Please, review your conclusion. You are supposed to answer your objectives, and some parts seems we are reading result again. I am sorry.

Comments on the Quality of English Language

The English could be improved to more clearly express the research.

Author Response

Dear Reviewer,

Thank you very much for your thorough review and constructive comments on our manuscript entitled “Characterization of the cellulose from Gagome kelp and its effect on the dough, gluten and starch as novel bread improvers” (Manuscript ID: 3505315). We greatly appreciate your time and effort in helping us improve the quality of our work. Below, we have addressed each of your comments point by point, and all changes made to the manuscript are highlighted in yellow for easy reference. Furthermore, in order to improve the quality of English language in our manuscript, we have send it for editing by a professional English Editing company.

General Remarks

After reading the manuscript , I realized that the manuscript showed in some parts the scientific rigour wanted, but in other parts I have missed it. The authors have presented critical evaluation only in some paragraphs. The references are not exactly current, besides the some sections should be reviewed. Thats why I have written some suggestions below in an attempt to improve the paper.

  1. The references are not exactly current, besides the some sections should be reviewed.

We apologize for this oversight. We have updated the references to include more recent and relevant studies, particularly in the Introduction and Discussion sections. Additionally, we have carefully reviewed and revised the sections flagged by you to improve clarity and coherence. The updated references are highlighted in yellow.

  1. Review the journal's citation rules. It looks like it's in superscript.

We apologize for this oversight. We have reformatted the citations to comply with the journal’s guidelines, ensuring they are no longer in superscript format.

Specific Comments

  1. Line 31: "bizarre" - I think this term very strong. Those of us who work in the field of developing new foods for special purposes do a lot of preliminary tests, and the odd treatments are discarded from the very first trials. Only treatments with potential continue to be assessed.

We apologize for this oversight. We agree that the term “bizarre” is too strong and have replaced it with “unconventional” to better reflect the preliminary nature of the tests.

Line 29: “ Unexpected appearance and texture may generate with the adding of healthier ingredients to bread formulation, and resulting in a unconventional product characteristics.”

  1. Line 37: "Therefore, the development of novel functional formulation on bread without quality and texture deficiency was proposed as a nutrition-improving strategy." - I am sorry, but I did not understand your point and I completely disagree if I understood.

We apologize for the lack of clarity. This sentence has been rephrased to: “Therefore, the present study proposes the development of a novel functional bread formulation that maintains quality and texture while enhancing nutritional value, and offering a viable strategy for improving the health benefits of bread.”

Line 56: “Thus, a worthwhile endeavor would be to clarify the potential effects of algal cellulose as a dietary fiber source in the human diet, especially for sensory characterization of food added algal cellulose.”

  1. Line 47: "Ingredients has been applied in cookies and bread" - This sentence needs authors.

We apologize for this oversight. We have corrected the sentence to: “Therefore, wide array of cellulose ingredients has been applied in cookies and bread, with the purpose of increasing the fiber content or reducing the calorific value[12].”

Line 48:“Therefore, wide array of cellulose ingredients has been applied in cookies and bread, with the purpose of increasing the fiber content or reducing the calorific value[12].”

  1. Line 56: "texture and sensory characterization" – Texture can be a sensory attribute as well.

We apologize for this oversight. The content of “texture” has been deleted.

Line 57: “Thus, a worthwhile endeavor would be to clarify the potential effects of algal cellulose as a dietary fiber source in the human diet, especially for sensory characterization of food added algal cellulose.”

  1. Line 74: Scientific name in Italics, please.

We apologize for this oversight. The scientific name has been italicized as requested.

Line 71:“The Gagome kelp (Kjellmaniella crassifolia), cultivated in the Yellow Sea near Dalian, China and harvested in May 2020, was provided by a local seaweed farm. ”

  1. Line 78: "The MCC" – Define acronym in the first moment.

We apologize for this oversight. We have defined “MCC” upon its first mention: “microcrystalline cellulose (MCC).”

Line 75:“The microcrystalline cellulose (MCC) was purchased from Sigma Co., Ltd.”

  1. Line 97: I think it's important to include your “room temperature”, It depends on the season and the country in which the experiment was conducted.

We deeply appreciate to provide valuable suggestions for improvement.We have specified the room temperature as “25°C” to provide clarity and context.

Line 95:“For the determination of water holding capacity, 40 mL deionized water was added into 1.0 g cellulose sample, the mixture was stored for 24 h at 25°C. ”

  1. Line 102: Perhaps you should type ma; mb; mau (L.121 and 127) in italics to be different.

We apologize for this oversight. The variables ma, mb, and mau in all manuscript have been italicized for consistency. The changes made to the manuscript are highlighted in yellow.

  1. Line 299: Hardness OK, but what about the other parameters? Were they calculated?

We apologize for the lack of clarity. The other TPA parameters of the samples, such as adhesion, cohesiveness, springiness, adhesiveness and chewiness were calculated by the software (TLPro 1.13-002, China). We have added a brief explanation in 2.2.19.

Line 322: “The hardness, adhesion, cohesiveness, springiness, adhesiveness and chewiness of the samples were calculated by the software (TLPro 1.13-002, China).”

  1. Line 306: Was this analysis done only in triplicate as well? I don't think that's enough repetition for this analysis.

We apologize for the lack of clarity. The determination on texture profile of the cellulose bread was performed for 10 times. The content of “All measurements in the present study were performed thrice” was incorrect.

In addition, we acknowledge the limitations in our initial methodology. The determination on cholesterol, glucose adsorptive capacity, and α-amylase inhibition activity, and glucose dialysis retardation index in the study were performed in ten times. The determination on cation-exchange capacity of GC was performed in six times. We apologize for this oversight. The content of “All measurements in the present study were performed thrice” was wrong. We had ignored the different repetitions in each determination. We have rectified the description about the number of repetition in methods sections with yellow highlighted.

However, the determination on water and oil holding, unsaturated fat and saturated fat of GC were only performed thrice. To address this, we have increased the number of repetitions to ten for these analyses and included a detailed error margin discussion in results 3.1 and 3.2. Additionally, we have refined the instrumental methods to reduce variability.

  1. Line 307: For a sensory test a lot of relevant information was not included, I suggest reading papers and improving your article. Which sensory test was performed? How many sessions were conducted? What was the profile of the assessors? Were the analyses performed in sensory booths? Did the assessors receive water to rinse the taste buds? How many men/women ? Are the assessors usually trained for evaluated this product?

Thank you for your valuable feedback regarding the sensory evaluation section. We sincerely apologize for the lack of detailed information and have revised the paragraph to address your concerns. The updated section now reads:

Line 327: “The sensory evaluation of GC-bread was conducted using a 10-point descriptive profile analysis (Table S1 and S2), as described in Akyüz et al. The GC-breads samples named 1-5 were cut into 2 cm × 2 cm × 2 cm pieces before given to the assessors. The test was performed in three sessions by a trained panel of ten assessors (five females and five males, aged 25–50 years) with prior experience in evaluating bakery products. Each session was conducted in standardized sensory booths under controlled conditions. Assessors were provided with water to cleanse their palates between samples. Protocols for the sensory evaluation were approved by the Experimental Ethics Committee of Dalian Ocean University (ethical approval No. 2024102102, approval date: October 21, 2024) and complied with the guidelines. Informed consent was obtained from all participants prior to the study.”

  1. Line 310: "Attributes agreed by the assessors are listed with a sensory analysis form of 15 scaled descriptive profile analysis form[23]. The results don't show 15. It's confusing here. Make it clearer, please. What scale did you use, please?

Thank you for your valuable feedback regarding the sensory evaluation section. We sincerely apologize for the lack of detailed information and have revised the paragraph to address your concerns. We have clarified that the 10-point scale was used to rate attributes such as color, aroma, tactility, state, shape, texture and acceptability. The results are presented as an average score. The 10-point descriptive profile analysis from have been uploaded in supplemental file (Table S1 and S2). In addition, since sensory experiments involve human, we have added institutional review board statement and informed consent statement in the back matter of the manuscript via the Foods editorial office.

  1. Line 327: 1%, 3%, 5% and 7% - It was confusing sometimes, this different percentages for texture and sensory in some analyses and storage times in others. I suggest that the research group sit down for a discussion and re-evaluate the priorities and the real objective of the study. Sometimes less is more.

Thank you for your valuable feedback. The description of “1%, 3%, 5% and 7%” represent as the contents of cellulose added in bread, as described in 2.2.17. Furthermore, the cold-stored time of GC-bread was “1, 3, 5 and 7 days”. We sincerely apologize for the confusing description. For the clear description, the content of “cellulose added in bread” was deleted, and the content of “cold-stored time” was replaced to “24, 72, 120 and 168 h” in 2.2.21.

Line 348: “The GC-bread was stored for 24, 72, 120 and 168 h at 4℃, and then the same TPA was performed. ”

  1. Line 353: Standardize the Material and methods is “fat” and the Results is “lipid”.

We apologize for the lack of clarity.We have standardized the terminology to use “fat” throughout the manuscript.

Line 373:“3.2 Interaction between the GC and fat

The fat-lowering effect of the cellulose has been widely reported in recent years. For the illumination of interaction between the GC and fat the adsorptive capacity of the GC on unsaturated fat and saturated fat was evaluated.”

  1. I'd ask you to review the results in table 1, I'm a little concerned about some of the results that showed a statistically significant difference and another point, did they all show a difference for the GC? It is quite strange.

We apologize for the lack of clarity.We have re-analyzed the data in Table 1 and clarified the statistical differences. The difference was only between the same characterization of MCC and GC. We sincerely apologize for the confusing description (p<0.05). We have added a brief explanation below the Table 1

Line 371: “The mark “*” in the table indicate a significant difference (p < 0.05) between the MCC and GC.”

  1. Line 426: I'm sorry, but I didn't understand your statistical analysis in Figure 1A and B. Are you comparing treatments and time periods? It doesn't seem appropriate, authors.

We apologize for the lack of clarity. We have re-analyzed the data in Figure 1A and 1B, and clarified the statistical differences. In addition,we have added a brief explanation below the Figure 1A and 1B.

Line 421: “The letters in the figure indicate a significant difference (p<0.05, n=10).”

  1. Line 427: This value (7501000) is quite strange in minute. Besides, it trespasses into the space of the previous interval on the graph

We apologize for the lack of clarity. The value of “time” (X axis) in Figure 1C has been standardized into “hours” and adjusted in the graph to avoid overlap.

  1. Line 447: What is the legend here for gray and green? I suggest using the same colors throughout the paper for GC, MCC and control.

 We sincerely apologize for the lack of detailed information in Figure 2B. The Figure 2B was a double Y-axis plot. The legend for green was “DPPH radical scavenging capacity (%)”, and for gray was “Hydroxyl radical scavenging capacity (%) ”. We have added the figure legend in Figure 2B. However, because of the double Y-axis plots in Figure 2B and Figure 3C with three experimental groups. the same colors throughout the paper for GC, MCC and control were difficult. We appreciate your thorough review and valuable feedback regarding our manuscript. We will take your suggestion into consideration in our further research.

  1. Line 477: What about MCC? Are you comparing treatments and time periods? It doesn't seem appropriate, authors.

We apologize for the lack of clarity. The results of MCC were missing.We have added a detailed comparison of MCC in Table 2, and the results and discussion of the effect on MCC were updated.

Line 496: “In this study, no significant (p<0.05) effect of the GC and MCC on color of dough could be observed in Table 2. The light values of GC-dough was higher than those of blank control and MCC. ”

  1. Line 495: MCC ? Review stastistics approach, please.

We apologize for the lack of clarity. The results of MCC were missing.We have added a detailed comparison of MCC in Figure 3, and the results and discussion of the effect on MCC were updated. In addition, we have re-evaluated the statistical approach and updated the results accordingly.

Line 476: “For the MCC-dough, the significant (p<0.05) increase on volume could be observed between 20 to 40 min.Similar results were supported by dough characteristics and bread quality with psyllium husk fiber. In such study, the values of pH indicate that the crumb of the control formulation recorded a significantly higher pH. ”

Line 487: “The effect of GC on pH of GC-dough was shown in Figure 3B. Although reduce on pH value could be observed on GC-dough, there was no significant (p<0.05) difference on dough either with or without GC and MCC in 20-60 min proofing. Such results might attribute to the esters, which could neutralized the lactic acid in dough proofing, and buffering the pH proofed GC-dough and MCC-dough. Moreover, such buffering capacity of the GC and MCC might be related to the high cation-exchange capacity shown in Figure 2A.”

  1. Line 518: 15.58±0.21 B in superscript.

We apologize for the lack of clarity. The value has been formatted as superscript. The changes in Table 3 were highlighted with yellow.

  1. Glutenin / Gliadin - (%) as well ? Is is missed. If they are all % insert them only once in the “Content” cell and the table will be cleaner.

We appreciate your thorough review and valuable feedback regarding our manuscript. The values in Table 3 were the quotient of Glutenin / Gliadin, which represent the content of glutenin in GC-gluten was 2.34±0.25 times higher than that of B. Considering that such description might cause confusion for reading, we have unified the values in Table 3 as a percentage statement of Gliadin / Glutenin. The changes in Table 3 were highlighted with yellow.

  1. Line 536: We've been thinking along the lines of reading, but in the end, with the texture and sensory analyses, which would be the cherry on the cake, especially since the product is bread, we no longer have the MCC to compare with and the study proposal changes completely. That's a pity.

Thank you for your thoughtful and constructive feedback regarding the comparison between GC and MCC in the context of texture and sensory analyses. We sincerely appreciate your suggestion to include MCC as a comparative control, which would indeed provide valuable insights into the functional properties of GC. However, after careful consideration and consultation with the MCC supplier (Sigma Co., Ltd, America), we encountered ethical limitations that prevented us from incorporating MCC into certain experiments, particularly those involving human sensory evaluation. 

The supplier confirmed that while MCC is widely used in scientific research, there is no documented data or literature supporting its safety for human trials. Given the ethical implications and the primary objective of our study—to explore the potential of GC as a natural bread improver—we decided to limit the use of MCC to scientific experiments investigating its physicochemical properties and effects on dough, gluten, and starch. In the production of fiber-enriched bread, we focused on evaluating the impact of different GC concentrations on bread quality, aiming to highlight its potential as a functional ingredient. Consequently, MCC was excluded from the sensory evaluation and subsequent analyses, including SEM observation, texture profile analysis, and cold-storage hardness determination, to ensure ethical compliance and alignment with the study objectives. 

We acknowledge that the absence of MCC in these analyses may limit the comparative scope of the study. However, we believe that the current results, which demonstrate the significant functional and structural contributions of GC to bread quality, provide a solid foundation for its potential application in bakery products. Your suggestion has greatly inspired us, and we will consider incorporating MCC in future studies, provided that sufficient safety data becomes available. 

Thank you for your understanding and constructive feedback. We hope the revised manuscript addresses your concerns and meets the journal standards. Please do not hesitate to contact us if further revisions are needed.

  1. Line 584: Where did you get these sensory attributes from? “Tactitlity” and ‘state’ for example. You had an explanatory session with the assessors about the attributes. Make that clear.

Thank you for your valuable feedback regarding the sensory attributes mentioned in the manuscript. The sensory evaluation was performed in three sessions by a trained panel of ten assessors with prior experience in evaluating bakery products. Each session was conducted in standardized sensory booths under controlled conditions. The sensory attributes, including ‘tactility’ and ‘state,’ were derived from an explanatory session with the trained assessors prior to the evaluation. During this session, the panelists were introduced to the specific attributes and their definitions to ensure consistent and accurate assessment. This approach aligns with established sensory evaluation protocols and ensures the reliability of the results. In addition, an 10-point descriptive profile analysis from have been uploaded in supplemental file (Table S1 and S2).

  1. Line 633: "GC were significantly (p<0.05)" Thit takes us back to the result, not the conclusion.

Thank you for your insightful feedback regarding the conclusion section. We agree that the inclusion of specific statistical results detracts from the purpose of the conclusion, which should focus on summarizing key findings and their implications rather than reiterating results. We have revised the conclusion to address this issue, and the updated sentences now reads:

Line 664: “In conclusion, this study demonstrates that Gagome kelp cellulose (GC) exhibits superior functional properties, including enhanced water and oil holding capacity, cholesterol adsorption, and glucose diffusion inhibition, compared to microcrystalline cellulose (MCC). ”

  1. Line 641: "was significantly (p<0.05)" - Thit takes us back to the result, not the conclusion.

Thank you for your constructive feedback regarding the conclusion section. We acknowledge that the inclusion of specific statistical results is more appropriate for the results section rather than the conclusion. To address this, we have revised the sentences to focus on summarizing the findings without reiterating statistical details. The updated paragraph now reads:

Line 667: “GC also significantly influences the properties of dough, gluten, starch, and bread, improving texture, glutenin content, and cold-storage stability. ”

  1. Please, review your conclusion. You are supposed to answer your objectives, and some parts seems we are reading result again. I am sorry.

Thank you for your valuable feedback regarding the conclusion section. We sincerely apologize for the redundancy and lack of focus on addressing the research objectives in the original conclusion. We have thoroughly revised the conclusion to ensure it summarizes the key findings and aligns with the objectives, avoiding repetition of results. The updated conclusion now reads: 

Conclusions:“In conclusion, this study demonstrates that Gagome kelp cellulose (GC) exhibits superior functional properties, including enhanced water and oil holding capacity, cholesterol adsorption, and glucose diffusion inhibition, compared to microcrystalline cellulose (MCC). GC also significantly influences the properties of dough, gluten, starch, and bread, improving texture, glutenin content, and cold-storage stability. Notably, bread containing 1% (m/m) GC showed optimal sensory characteristics and storage performance, highlighting its potential as a natural bread improver. These findings suggest that GC is a promising biocompatible ingredient for developing functional bread formulations, offering a sustainable strategy for enhancing bakery products. Future studies should focus on scaling up production and evaluating consumer acceptance to further validate its commercial applicability.” 

Once again, we deeply appreciate your insightful comments, which have significantly improved the quality of our manuscript. We hope that the revised version meets your expectations and the standards of Foods. Please do not hesitate to contact us if further revisions are needed.

Sincerely,
Xiang Li,
On behalf of all authors

Round 2

Reviewer 1 Report

Comments and Suggestions for Authors

Quality of the research is significantly improved

Author Response

Dear Reviewer,

Thank you for your positive feedback and for acknowledging the significant improvements in our manuscript entitled “Characterization of the cellulose from Gagome kelp and its effect on the dough, gluten and starch as novel bread improvers” (Manuscript ID: 3505315). We are deeply encouraged by your recognition of the enhanced research quality, which reflects the invaluable guidance provided during the review process. Your constructive critiques were pivotal in refining the experimental design, strengthening the statistical analyses, and clarifying the study’s contributions to the field of natural bread improvers.

We sincerely appreciate the time and expertise you dedicated to evaluating our work. Your insights have not only elevated the current study but also inspired future research directions, particularly in exploring marine-derived functional ingredients for sustainable food applications.

Thank you once again for your support and for contributing to the advancement of this research.

Sincerely,
Xiang Li,
On behalf of all authors

Reviewer 2 Report

Comments and Suggestions for Authors After another evaluation of the manuscript, I see a great improvement in the quality of the paper. The authors have accepted some of my requests. They added more authors to better substantiate the methodology and corrected tables and graphs. 

English is always useful to ask a native speaker for a final appreciation.

Authors, you still have room to improve the paper.

L.51- " Stanford et al[14]. " - Not correct . You' need to check all the references.

L.99, 110, 125, 141- Check the use of preposition "in" . Check the whole Material and Methods.

L.317- Which sensory test was performed?- It is important to be described in Material and Methods.

L.410- In my opinion, Figure 1 is still not adequate if you want to compare GC and MCC.
If you are statistically comparing glucose and time and glucose and concentration. That's why, from the first version, I signaled the research group to discuss and re-evaluate the objectives of the study. Think about whether this 30, 60 and 90 time is compromising the analysis. 

L. Authors, you still have problems with Figure 3B, do you use #, *, + 
Is that usual?  Sorry, but wouldn't it be better to present these results in a table? We compare treatments, OK But what about time 20, 40, 60?
Shouldn't this statistical analysis be a two-way ANOVA? 
It's your call, obviously the authors decide how they want to evaluate their data, I'm just trying to save you from reproaches in the future.

Answer 23-  Please, you should include your answer  in "Study Limitations" since many researches will wonder the same thing I did.

"Thank you for your thoughtful and constructive feedback regarding the comparison between GC and MCC in the context of texture and sensory analyses. We sincerely appreciate your suggestion to include MCC as a comparative control, which would indeed provide valuable insights into the functional properties of GC. However, after careful consideration and consultation with the MCC supplier (Sigma Co., Ltd, America), we encountered ethical limitations that prevented us from incorporating MCC into certain experiments, particularly those involving human sensory evaluation." 

L.491- What about comparing means among treatments? Lower-case letters ?

L.600- It's a bit strange that the spider graph starts at 6.5. Make the scale more spaced out, but start from the real number.

L.638- What about comparing means among treatments? Lower-case letters ?

Author Response

Dear Reviewer,

Thank you very much for your thorough review and constructive comments on our manuscript entitled “Characterization of the cellulose from Gagome kelp and its effect on the dough, gluten and starch as novel bread improvers” (Manuscript ID: 3505315). We greatly appreciate your time and effort in helping us improve the quality of our work. Below, we have addressed each of your comments point by point, and all changes made to the manuscript are highlighted in yellow for easy reference.

General Remarks

After another evaluation of the manuscript, I see a great improvement in the quality of the paper. The authors have accepted some of my requests. They added more authors to better substantiate the methodology and corrected tables and graphs.

English is always useful to ask a native speaker for a final appreciation.

Authors, you still have room to improve the paper.

  1. English is always useful to ask a native speaker for a final appreciation.

   Thank you very much for your thorough review and constructive comments on our manuscript. We have sought the assistance of a native English speaker to further refine the language and ensure the manuscript meets the highest standards of readability and professionalism. All changes made to the manuscript are highlighted in yellow for easy reference.

Specific Comments

  1. L.51- " Stanford et al[14]. " - Not correct . You' need to check all the references.

   We apologize for the oversight. The reference formatting has been corrected to “Stanford [14].” We have also carefully reviewed all references to ensure consistency with the journal’s guidelines. 

Line 52: “As early as 1885, Stanford [14] had proposed the concept of algal cellulose, which is isolated from the insoluble residue after the extraction of alginates from brown algae.”

  1. L.99, 110, 125, 141- Check the use of preposition "in" . Check the whole Material and Methods.

   We apologize for this oversight. We have reviewed the use of the preposition “in” throughout the Materials and Methods section and made necessary corrections to ensure grammatical accuracy and clarity. In addition, we have sought the assistance of a native English speaker to further refine the language and ensure the manuscript meets the highest standards of readability and professionalism. All changes made to the manuscript are highlighted in yellow for easy reference.

  1. L.317- Which sensory test was performed?- It is important to be described in Material and Methods.

Thank you for your valuable feedback regarding the description of the sensory test in the Materials and Methods section. We have revised the paragraph to provide a more detailed explanation of the sensory evaluation methodology, including the type of sensory test performed and its alignment with MDPI guidelines.

Line 316: “2.2.20 Sensory test description of the GC-bread

A quantitative descriptive analysis (QDA) of GC-bread was tested using a 10-point descriptive profile analysis (Table S1 and S2), as described in Akyüz et al. ”

  1. L.410- In my opinion, Figure 1 is still not adequate if you want to compare GC and MCC. If you are statistically comparing glucose and time and glucose and concentration. That's why, from the first version, I signaled the research group to discuss and re-evaluate the objectives of the study. Think about whether this 30, 60 and 90 time is compromising the analysis.

Thank you for your insightful feedback regarding Figure 1 and the statistical comparison between GC and MCC.

The primary objective of this study was to evaluate the functional properties of GC as a novel bread improver, with MCC serving as a reference. To achieve this, we designed the experiment to address two independent questions:

Objective 1: Compare the effects of different treatments (GC vs. MCC) under identical conditions.

Objective 2: Compare the effects of different experimental variables (concentration of glucose /time) within the same treatment group (GC or MCC).

The time points (30, 60, and 90 minutes) were selected based on preliminary experiments to capture the dynamic behavior of glucose adsorption and diffusion. These points were not intended to compare interactions between treatments and variables but rather to evaluate trends within each treatment group. Therefore, given the independent nature of these comparisons, we employed two separate one-way ANOVA analyses to: (1) Assess the significance of differences between GC and MCC at each time point; (2) Evaluate the significance of changes over time within each treatment group. This approach aligns with the goal of our study to isolate the effects of GC and MCC independently, rather than examining interactions between treatments and variables.

Above all, to improve clarity, we have revised Figure 1 to include a revised caption explaining the independent comparisons and their relevance to the study objectives.

Line 411: “The capital letters in the table indicate a significant difference (p<0.05) with treatment, while the lowercase indicate a significant difference (p<0.05) with concentration of glucose (A) and time (B).”

Once again, we deeply appreciate your insightful comments, which have significantly improved the quality of our statistical analysis.

  1. L. Authors, you still have problems with Figure 3B, do you use #, *, + Is that usual? Sorry, but wouldn't it be better to present these results in a table? We compare treatments, OK But what about time 20, 40, 60? Shouldn't this statistical analysis be a two-way ANOVA? It's your call, obviously the authors decide how they want to evaluate their data, I'm just trying to save you from reproaches in the future.

We agree that the use of multiple symbols (#, *, +) in Figure 3B may be confusing. We have reorganized the data into a table format (Table 2) for better clarity, and the relative paragraph were all updated.

Similar to Answer 4, given the independent nature of these comparisons, we employed two separate one-way ANOVA analyses of Table 2. Furthermore, to improve clarity, we have revised Table 2 to include a revised caption explaining the independent comparisons and their relevance to the study objectives.

Line 489: “The capital letters in the table indicate a significant difference (p<0.05) with treatment, while the lowercase indicate a significant difference (p<0.05) with time.”

  1. Answer 23- Please, you should include your answer in "Study Limitations" since many researches will wonder the same thing I did.

"Thank you for your thoughtful and constructive feedback regarding the comparison between GC and MCC in the context of texture and sensory analyses. We sincerely appreciate your suggestion to include MCC as a comparative control, which would indeed provide valuable insights into the functional properties of GC. However, after careful consideration and consultation with the MCC supplier (Sigma Co., Ltd, America), we encountered ethical limitations that prevented us from incorporating MCC into certain experiments, particularly those involving human sensory evaluation."

Thank you for your valuable feedback regarding the "Study Limitations". We have incorporated our response regarding the ethical limitations of using MCC in human sensory evaluation before the “Conclusions”. This addition provides context for future researchers and addresses potential questions about the experimental design. 

Line 646: “This study has the following limitations. First, ethical restrictions prevented the use of MCC in human sensory evaluation experiments, which may have constrained direct comparative analyses with GC. Second, the relatively limited sample size could affect the generalizability of the findings. Future studies should prioritize expanding the sample size and investigating functional properties of alternative cellulose sources, particularly those derived from marine biomass, to further validate and broaden the applicability of these results in bakery product development.”

  1. L.491- What about comparing means among treatments? Lower-case letters ?

We apologize for the oversight. Similar to Answer 5, given the independent nature of these comparisons, we employed two separate one-way ANOVA analyses on the color of the GC-dough (Table 2). Furthermore, to improve clarity, we have revised Table 2 to include a revised caption explaining the independent comparisons and their relevance to the study objectives.

Line 489: “The capital letters in the table indicate a significant difference (p<0.05) with treatment, while the lowercase indicate a significant difference (p<0.05) with time.”

  1. L.600- It's a bit strange that the spider graph starts at 6.5. Make the scale more spaced out, but start from the real number.

   We have adjusted the spider graph to start from the actual baseline value (0) and spaced out the scale for better readability. This change ensures the graph accurately represents the data. 

  1. L.638- What about comparing means among treatments? Lower-case letters ?

   We apologize for the oversight. We have updated the statistical analysis to include lowercase letters for comparing means among treatments in Table 5, improving the clarity of the results. 

Once again, we deeply appreciate your insightful comments, which have significantly improved the quality of our manuscript. We hope that the revised version meets your expectations and the standards of Foods. Please do not hesitate to contact us if further revisions are needed.

Sincerely,
Xiang Li,
On behalf of all authors
